# “Bioinspired” Membrane-Coated Nanosystems in Cancer Theranostics: A Comprehensive Review

**DOI:** 10.3390/pharmaceutics15061677

**Published:** 2023-06-08

**Authors:** Nimeet Desai, Dhwani Rana, Shreya Pande, Sagar Salave, Jyotsnendu Giri, Derajram Benival, Nagavendra Kommineni

**Affiliations:** 1Department of Biomedical Engineering, Indian Institute of Technology Hyderabad, Kandi 502285, India; bm21resch11003@iith.ac.in (N.D.); jgiri@bme.iith.ac.in (J.G.); 2National Institute of Pharmaceutical Education and Research (NIPER), Ahmedabad 382355, India; dhwanirana73@gmail.com (D.R.); sagarsalave1994@gmail.com (S.S.); derajram@niperahm.res.in (D.B.); 3Center for Biomedical Research, Population Council, New York, NY 10065, USA

**Keywords:** cancer, theranostics, nanoparticles, cell membrane, membrane coating, tumor targeting

## Abstract

Achieving precise cancer theranostics necessitates the rational design of smart nanosystems that ensure high biological safety and minimize non-specific interactions with normal tissues. In this regard, “bioinspired” membrane-coated nanosystems have emerged as a promising approach, providing a versatile platform for the development of next-generation smart nanosystems. This review article presents an in-depth investigation into the potential of these nanosystems for targeted cancer theranostics, encompassing key aspects such as cell membrane sources, isolation techniques, nanoparticle core selection, approaches for coating nanoparticle cores with the cell membrane, and characterization methods. Moreover, this review underscores strategies employed to enhance the multi-functionality of these nanosystems, including lipid insertion, membrane hybridization, metabolic engineering, and genetic modification. Additionally, the applications of these bioinspired nanosystems in cancer diagnosis and therapeutics are discussed, along with the recent advances in this field. Through a comprehensive exploration of membrane-coated nanosystems, this review provides valuable insights into their potential for precise cancer theranostics.

## 1. Introduction

Cancer, a multifaceted disease with a high mortality rate, poses a significant burden on global healthcare systems. Achieving an accurate diagnosis and effective therapy presents a major challenge in cancer treatment. The integration of theranostics, which combines diagnostic and therapeutic approaches, has emerged as a promising trend to address this challenge. Theranostics aims to streamline procedures, minimize treatment delays, and improve patient care by offering enhanced diagnostic capabilities, targeted drug delivery to tumors, and reduced toxicity to healthy tissues. Utilizing theranostic nanomedicines effectively can significantly contribute to achieving these objectives. Nanotechnology has paved the way for targeted and efficient cancer diagnosis and therapy by enabling the delivery of drugs and imaging agents to tumor sites. Various nanomaterials offer opportunities for the development of nanoparticle-based modalities for therapeutic, preventive, and detection purposes, yielding highly effective results [1,2]. As nano-imaging and nano-therapy advance, novel opportunities arise for more efficient cancer treatments, making nanotechnology integration a promising pathway [3].

Nanoparticle targeting plays a crucial role in anticancer therapy by enhancing the efficacy and specificity of drug delivery to cancerous cells while minimizing damage to healthy tissues. The need for nanoparticle targeting arises from the limitations of conventional chemotherapy, which often suffers from low selectivity, poor pharmacokinetics, and systemic toxicity. There are two main approaches to nanoparticle targeting: passive targeting and active targeting [4]. Passive targeting involves taking advantage of the unique characteristics of tumor vasculature and the enhanced permeability and retention (EPR) effect. The EPR effect refers to the abnormal leakiness and disorganized nature of blood vessels in tumors, which allows nanoparticles to selectively accumulate in tumor tissues due to their larger size and prolonged circulation time. This effect is primarily driven by the impaired lymphatic drainage in tumors. Passive accumulation emerged as a fundamental strategy for targeting tumors in the initial stages of the “nanomedicine” era. This approach gained significant attention and recognition because of its straightforwardness in achieving efficient delivery to tumor sites by manipulating factors such as nanoparticle size, shape, and surface charge. Consequently, it has formed the foundation for numerous clinically available formulations based on liposomal or albumin nanoparticles [5].

However, a comprehensive analysis of the scientific literature between 2005 and 2015, incorporating 232 data sets, revealed that the passive delivery efficiency of systemically administered nanoparticles to solid tumors is remarkably low, with a median value of only 0.7% [6]. According to a multivariate analysis of relevant parameters, tumor type, tumor model, and nanomaterial properties are the main determinants of passive accumulation efficiency. It was observed that the high interstitial fluid pressure within tumor tissues hinders the extravasation of nanoparticles. Moreover, certain nanoparticles that manage to enter the intercellular space of tumors through the EPR effect may be forced back into the bloodstream due to the elevated fluid pressure within the tumor interstitium [7]. Notably, blood cancers, early stage tumors, and small metastasized cancers either lack the EPR effect or exhibit only minimal levels. Furthermore, the EPR effect is significantly limited or absent in certain cancer types and even within different regions of the same tumor, owing to tumor heterogeneity [8]. It is due to these critical pitfalls that the active targeting approach has taken center stage in academic research. It involves modifying the nanoparticles’ surface with ligands, antibodies, aptamers, or peptides that can specifically recognize and bind to receptors overexpressed on the surface of tumor cells or tumor-associated vasculature [9]. This active targeting approach aims to enhance the selectivity and affinity of nanoparticles for cancer cells, thereby improving their cellular uptake and therapeutic efficacy. Active targeting ligands can be chosen based on the specific molecular markers expressed on tumor cells, such as growth factor receptors, integrins, or other specific receptors associated with tumor progression. By actively targeting these receptors, nanoparticles can be directed to the tumor site, leading to increased internalization into cancer cells and reducing off-target effects [10].

In recent years, the concept of “bioinspired” functionalization of nanoparticles has emerged as a prominent area of research, drawing considerable attention. This innovative approach involves the utilization of cell-derived membranes to coat nanoparticles, offering distinct advantages over conventional active targeting strategies. An important advantage lies in the inherent biocompatibility and safety provided by endogenous cell membranes. Coating nanoparticles with these membranes enables prolonged circulation within the body without triggering an immune response. Furthermore, this approach shows promise in addressing the challenge of cancer heterogeneity by allowing the design of personalized treatments through the choice of cell source [11]. Remarkably, this functionalization strategy eliminates the need for complex surface modifications typically associated with other active targeting approaches. Different cell membranes can confer unique functions on the resulting nanosystem, leading to diverse in vivo behaviors. While membrane coating achieves targeting effects similar to conventional strategies, it offers additional benefits due to its multifunctional properties [12]. In contrast, nanoparticles functionalized with ligands or aptamers are prone to immune recognition and clearance by the mononuclear phagocytic system, posing challenges during systemic circulation. Membrane-coated nanosystems, on the other hand, possess inherent immune evasion capabilities. They experience minimal protein corona formation and do not suffer from loss of targeting ability as observed with covalently conjugated ligands [13]. Moreover, the use of membranes derived from activated immune cells or bacteria can serve secondary functions by modulating immune activity within the tumor microenvironment. Cell membrane coating presents a universal functionalization approach that circumvents the need for chemical processes, allowing researchers and clinicians to select the most suitable nanosystem for drug encapsulation. This approach holds tremendous potential for advancing drug delivery systems in a safe, efficient, and versatile manner [14].

This review article thoroughly investigates the potential of bioinspired membrane-coated nanosystems for targeted cancer diagnosis and therapy. It explores various aspects, including cell membrane sources, isolation techniques, nanoparticle core selection, coating approaches, and characterization methods. Moreover, it highlights approaches to increasing the multi-functionality of these nanosystems through lipid insertion, membrane hybridization, metabolic engineering, and genetic modification. This review discusses the applications of bioinspired nanosystems in cancer diagnosis and therapeutics, along with recent advances in this field. By providing a comprehensive exploration of membrane-coated nanosystems, this review aims to contribute to the advancement of precise cancer theranostics.

## 2. Membrane Sources and Attributes

Intravenous administration of image contrast agents/drugs typically results in their passage through the circulatory system, followed by renal elimination, which can affect the kinetic parameters within the body. Incorporating these agents within nanoparticles (NPs) can enhance their pharmacokinetic performance and allow for targeted delivery [15]. However, several biological barriers, such as immune recognition/activation and systemic clearance, must be overcome. PEGylation, charge modulation (by attaching zwitterionic entities), or utilization of surface ligands that bypass immune surveillance are some of the available surface modification approaches to improve NP performance [16,17,18]. However, in most cases, the benefits of employing such strategies are undermined by the added complexity of synthesis. Hence, the attention of the scientific fraternity has shifted toward cell membrane coating as a “bioinspired” strategy to functionalize NPs for biomedical applications. Depending on the membrane source, the corresponding nanosystem will typically have a unique set of properties that can be leveraged for theranostics applications. Figure 1 represents various sources for procuring functional membranes whereas an overview of membrane sources and their attributes are enlisted in Table 1. The following section discusses various classes of cell membrane sources along with their key attributes.

### 2.1. Blood Cells

Blood cells are a highly diverse and extensively utilized class of membrane source, encompassing a variety of cell types produced in the bone marrow and circulating in the bloodstream. This includes erythrocytes (red blood cells; RBCs), platelets (PLTs), and leukocytes such as macrophages, dendritic cells (DCs), and natural killer (NK) cells.

Erythrocytes are the most abundant cellular component of blood, with approximately 5 million cells per microliter. Due to their size (7–8 μm diameter with a thickness of approximately 1 μm at the center) and lack of nuclei/sub-cellular organelles, their membrane is easily extractable and purifiable. Their long retention time of up to 120 days in humans makes them ideal for conferring long-circulation properties to entrapped NPs [19]. Furthermore, their surface is rich in “self-tagged” proteins such as CD47 (which protects them from phagocytosis by modulating recognition by the signal-regulatory protein alpha glycoprotein), as well as glycans and acidic sialic acid fractions. These attributes collectively provide immune evasion and deterrence from forming the protein corona [20]. Other membrane proteins such as C8 binding protein, homologous restriction protein, decay accelerating factor, membrane cofactor protein, and complement receptor 1 also play a role in resistance to complement system attacks [21]. It is widely accepted that Hu et al. (2011) reported the first bioinspired nanosystem comprising biodegradable polymeric NPs encapsulated by an erythrocyte membrane [22]. Since then, their abundance and ease of availability have led to their widespread use [23].

PLTs, which originate from mature megakaryocytes, are another type of anucleate blood cell with significant applications in developing bioinspired nanosystems [24]. They are abundant in the blood (approximately 150,000–350,000 cells/mL) and have a circulatory lifespan of 7–10 days. PLTs play a vital role in generating an inflammatory response and hemostasis, especially after thrombosis. Many studies have established a correlation between platelet-induced hemostatic properties and cancer progression/metastasis. PLTs can recognize, interact with, and cover circulating tumor cells (CTCs), allowing them to evade immune clearance and spread to new tissues [25]. The surface marker proteins of PLTs, including P-Selectin (CD62p), PECAM-1, CD47, and CD44 receptors, play a crucial role in this function. The unique properties of these surface marker proteins make PLT membranes an attractive coating material for NPs [26]. Coating with PLT membranes can confer beneficial characteristics such as immune evasion and specific adhesion to injured vessels and tumor tissues. Furthermore, the inflammatory characteristics of tumors can “recall” PLTs to accumulate passively at the site of cancerous growth [27].

Leukocytes are derived from multipotent hematopoietic stem cells in the bone marrow, where they differentiate and mature before entering the bloodstream. As immune cells, they play a vital role in protecting the body from infections, repairing tissue injuries, and resisting diseases by engulfing foreign invaders. Leukocytes have migration properties similar to CTCs and share adhesion molecules with the vascular endothelium, allowing for interaction with activated endothelial cells [28]. Leukocytes accumulate near the endothelial cell wall of blood vessels due to rheological differences with erythrocytes. To enter the target site, leukocytes undergo a “rolling adhesion phenomenon,” where weak interactions are mediated by selectins (on endothelial cells) and their ligands (on leukocytes), such as P-selectin glycoprotein ligand-1 and L-selectin. Later, they induce strong/firm adhesion mediated by the binding of the intercellular ICAM-1 adhesion molecule expressed on the vascular endothelium to leukocytes’ β2 integrins (e.g., lymphocyte function-associated antigen-1, macrophage-1 antigen) [29]. Additionally, several chemoattractants present in the tumor microenvironment (in the form of surface receptors of cancer cells or soluble chemokines) are imperative for leukocytes’ movement toward the tumor [30]. Unique features such as adhesive interactions with vascular walls and their active recruitment at tumor/inflammatory locations make them lucrative membrane sources. Some important receptors/adhesion molecules and their tumor-associated roles are discussed below:Macrophage: C-C chemokine receptor 2 (CCR2), vascular cell adhesion molecule-1 (VCAM-1), and intercellular adhesion molecule-1 (ICAM-1) facilitate the movement towards inflammatory tumor sites. α4 and β1 integrins interact with VCAM-1 on cancer cell membranes, allowing selective interaction with target cancer cells [31]. In addition, CD45, CD11a, and glycans act as functional molecules that aid in tumor localization by preventing internalization by phagocytes [32].DCs: They activate T cells by presenting antigens through their broad spectrum of membrane peptide/MHC complexes. ICAM-3, CD40, CD44, and integrins are a few molecules that aid in the adhesion and interaction of DCs [33]. DCs, as a membrane source, provide the advantage of lymph node targeting via the CCR7 receptor [34].NK cells: They play a crucial role in cancer elimination by monitoring the atypical expression of MHC-I and stress proteins on the cell surface. Despite the absence of tumor antigen-specific cell surface receptors, membranes sourced from NK cells possess several alternative receptors (such as NKG2D, NKp44, NKp46, NKp30, and DNAM-1) that enable them to recognize cancer cells, enhancing biocompatibility and tumor homing ability [35,36].

### 2.2. Cancer Cells

Cancer cells have the unique advantage of indefinite propagation, which means that they can continue to divide and proliferate indefinitely under certain conditions. This property is known as immortalization and is a hallmark of cancer cells. Cancer cells achieve immortalization through various genetic and epigenetic changes that allow them to bypass the normal regulatory mechanisms that control cell division and death [37]. For example, mutations in genes that regulate cell cycle checkpoints, such as p53 or RB1, can lead to uncontrolled cell proliferation and evasion of apoptosis, which are characteristic of cancer cells [38]. This indefinite propagation ability makes cancer cells an excellent source for cell membrane isolation because it allows for the continuous production of large quantities of cells for experiments. This is especially important for the isolation of cell membranes, which can be a labor-intensive and low-yield process. By using cancer cells, researchers can generate large amounts of membrane material from a single source, ensuring the consistency and reproducibility of their experiments [39].

The membrane properties of cancer cells are often distinct from those of healthy cells. These unique properties can include the overexpression of specific receptors or antigens as well as changes in lipid composition. These membrane features are significant contributors to the development, progression, and metastasis of tumors. CD47 overexpression on the cell membrane is known to contribute to immune evasion [40]. The mechanism of homotypic response in cancer cells heavily relies on cancer cell adhesion molecules (CCAMs). CCAMs comprise membrane receptors such as selectins, cadherins, integrins, the immunoglobulin superfamily (Ig-SF), and lymphocyte-homing receptors (e.g., CD44). Cadherins significantly impact cell–cell adhesion, signaling, migration, and gene regulation. Integrins, on the other hand, play a crucial role in cell–cell and cell-extracellular membrane interactions, which are essential for cell proliferation, differentiation, and migration [41]. In addition to these CCAM proteins, the Thomsen–Friedenreich glycoantigen (TF-Ag) associated with tumors, along with galectin-3, can mediate metastatic cell homotypic aggregation [42].

The presence of the abovementioned surface markers makes them an excellent choice for developing tumor-targeted nanosystems. Fang et al. were the first to report the functionalization of polymeric NPs with a layer of membrane coating obtained from cancer cells. The coating resulted in a 20-fold increase in particle uptake compared to non-coated systems due to homotypic binding mechanisms [43]. This discovery has led to extensive exploration of cancer cells for localized tumor theranostics. In addition to cancer cell lines, cancer cells can be sourced directly from a patient’s tumor biopsy or indirectly through patient-derived xenografts. This allows for the isolation of membrane coatings with patient-specific surface markers and tumor-associated antigens, which offer unique immunotherapeutic advantages [44].

### 2.3. Stem Cells

Stem cells are a type of progenitor cell that possess self-renewal and multi-lineage differentiation capabilities. Among them, mesenchymal stem cells (MSCs) are the most extensively studied for the functional coating of nanosystems [45]. These adult stem cells can be found in various tissues and organs, such as bone marrow, adipose tissue, and umbilical cord tissue, and have the capacity to differentiate into multiple cell types, such as osteoblasts, chondrocytes, adipocytes, and myocytes. Additionally, they interact with the immune system to regulate inflammation and immune responses. In the context of cancer, MSCs can directly influence tumor cells and promote the formation of tumor vasculature [46]. They may also differentiate into other types of cells in the tumor stroma, such as tumor-associated fibroblasts. Exogenous MSCs have an inherent tendency to migrate toward the microenvironment of developing tumors, attributed to their homing effect. The mechanism of MSCs’ homing to tumors is similar to the chemotaxis of immune cells to the site of inflammation [47]. During tumorigenesis, adhesion molecules, chemokines, and growth factors are overexpressed, inducing MSCs to become integral components of the tumor stroma. Ligands and corresponding receptors, such as SDF-1/CXCR4, PDGF/PDGFR, and VEGF/VEGFR, play important roles in this migration by binding to surface proteins on MSCs. The tumor tropism of MSCs is also influenced by the presence of membrane proteins such as TGF-β, E-selectins, and P-selectins [48]. Since MSCs hardly express MHC molecules, their targeting mechanism is tumor-specific rather than species-specific. This characteristic allows the use of MSCs from other species and expands the sources of cells for cancer targeting [49].

### 2.4. Extracellular Vesicles

Extracellular vesicles (EVs) are small, anucleated, and dynamic membranous particles present in the extracellular space, blood, and different body fluids. The primary structural components of the EVs are lipids, nucleic acids, and proteins associated with the plasma membrane and cytosol. EVs are mainly involved in intracellular communication as well as modulating important cellular processes such as homeostasis, regulation of inflammation, and promotion of tissue repair [50,51]. EVs have been successfully isolated from diverse sources, encompassing mammalian and prokaryotic cell cultures, blood plasma, bovine milk, and plants [52]. EVs are formed as a result of a complex cellular process that begins with the inward folding of the endosomal limiting membrane, forming the intraluminal vesicles (ILV), which then form a unique cellular compartment called multivesicular bodies (MVBs). MVBs later merge with the plasma membrane and are eventually secreted as exosomes. Based on the route of formation, structure, size, cargo profile, membrane compositions, and functions, EVs can be further divided into three different types, viz., exosomes (30–100 nm), microvesicles (MVs, 100–1000 nm), and apoptotic bodies (>1000 nm) [53]. Apoptotic bodies are non-living fragments released as a consequence of the rupture of cells due to apoptosis, containing histones and genomic DNA [54]. The process of microvesicle biogenesis entails the vertical transport of molecular cargo to the plasma membrane, a reorganization of membrane lipids, and the utilization of contractile machinery on the cell surface to facilitate vesicle formation through pinching [55]. Upon their release into the extracellular space and entry into circulation, these vesicles have the capacity to transfer their cargo to adjacent or distant cells, leading to phenotypical and functional alterations [56]. Among them, exosomes are extensively studied for their biological attributes and therapeutic applications. 

Exosomes are commonly detected in a diverse range of cell types, including tumor cells, mesenchymal stem cells, fibroblasts, neurons, endothelial cells (ECs), and epithelial cells, as reported in the scientific literature. Exosomes are able to interact with their intended targets because their membranes are enriched with transmembrane proteins such as tetraspanins (CD9, CD63, and CD81), antigen-presenting molecules (tumor-associated antigens), glycoproteins and adhesion molecules, ligands, and receptors. Due to the nature of their membrane, exosomes are not immunogenic, which prevents the immune system from recognizing them and lengthens the circulation half-life of the drug carrier system they are encapsulating [57]. The exosomes derived from mesenchymal stem cells (MSCs-EXs) express various markers such as CD9, CD63, CD55, CD59, CD81, Alix, and EP-CAM on their surface. CD9, CD63, Alix, and EP-CAM are the markers used for the isolation of exosomes. Owing to the presence of opsonin and coagulating factor-inhibiting markers (CD55 and CD59), the MSCs-EXs are uniformly distributed in the bloodstream. MSCs-EXs have shown the potential to inhibit tumor growth and immunomodulation. When the exosomal membrane-coated drug carriers are administered to the host, the circulation half-life is increased [58]. 

Tumor cells secrete a higher number of exosomes than healthy body cells. Exosomes from tumor cells contain similar membrane proteins to the tumor cells themselves, which makes them more likely to home in on the tumor. In addition to immunosuppressive proteins such as PD-L1 and death receptor ligands such as FasL and TRAIL, tumor-derived exosomes express tumor-associated antigens and MHC components that can help modulate the immune system and have anticancer effects. The transmembrane proteins on the surface of exosomes can bind directly to receptors on tumor cells, leading to the activation of apoptotic signaling pathways [59].

### 2.5. Viral Capsids

Viruses are tiny parasites that can only survive within host cells and are thought to have evolved alongside human genetic blueprints [60]. They consist of genetic material, capsid, envelope (in some viruses), and matrix proteins. The genetic material of viruses is delicate, contains either DNA or RNA but not both, and is surrounded by a shell made up of repeating protein units called capsids. The capsid, or envelope, of viruses contains various attachment proteins that help the virus invade the host cell [61,62]. There are two types of viral capsids: helical and icosahedral. The helical capsid forms when capsid proteins coil around the virus’s helical genome. The icosahedral capsid is made up of three identical or different proteins and is the structural unit of the capsid. The nucleocapsid, which is composed of only one type of protein along with the viral genetic material, requires less energy to assemble [63,64].

Various viral proteins are reported to be involved in altering and inhibiting tumor growth [65]. Viral capsids derived from cowpea mosaic virus, cowpea chlorotic mosaic virus, and MS2 bacteriophage are a few examples of commonly used viral capsids for tumor targeting. The MS2 capsid was used to deliver molecules to hepatic carcinoma cells upon modification with an HCC targeting peptide, which showed 104-fold higher affinity towards HCC than endothelial cells and hepatocytes [66]. Another protein, Rep6/U94, is a single-stranded DNA-binding, helicase-ATPase, helicase protein that is expressed during the replication of the virus and is involved in DNA replication. This protein has been evaluated for its antitumor effect in prostate cancer. It causes the upregulation of FN-1, which eventually reduces tumorigenesis [67]. Human viral proteins, such as Parvovirus NS1, are 672 amino acids (aa). When Thr-435 and Ser-473 residues are modified, this leads to tumor cell death via mitochondrial outer membrane permeabilization, DNA damage, cell cycle arrest, and caspase activation [68]. Similarly, in a protein, Rep78 from the Adeno-Associated Virus, the function of p5 is enhanced to block the cell cycle and cause DNA damage [69]. Plant viruses are non-infectious and safe without causing any biological response in humans as compared to most mammalian viruses. The icosahedral capsid of the plant virus JgCSMV was modified with folic acid to deliver an anticancer drug [70]. Virus-like NPs have received extensive attention because of their higher encapsulation efficiency, increased circulation time in the bloodstream, biocompatibility, and controlled release. Though immunogenicity is an important property when it comes to cancer therapy, using virus-like NPs as carriers for diagnostic molecules is required to protect them from the immune system. This can be achieved by attaching polyethylene glycol (PEG), which is famous for increasing circulation time and stability in plasma [71,72]. The viral capsids can also be easily modified, conjugated with other cell membranes, and complexed with a myriad of chemotherapeutic agents to enhance the anticancer effect [73,74].

### 2.6. Bacteria

The bacterial envelope is an essential organelle that surrounds the cytoplasm and maintains the shape and integrity of the cell while also regulating the exchange of nutrients, metabolites, and signaling molecules. It is composed of three main components: the cell wall, cell membrane, and associated proteins [75]. In gram-negative bacteria, the envelope consists of an outer membrane of lipopolysaccharides, an inner membrane of phospholipids, and a periplasmic space containing a peptidoglycan cell wall. Outer membrane vesicles (OMVs), small spherical structures ranging from 20–400 nm, are produced by all gram-negative bacteria. These vesicles form as a result of bulges in the outer membrane of the envelope that detach from the cell and are released into the surrounding environment [76,77,78]. OMVs can participate in horizontal gene transfer, cell-to-cell communication, and metabolite exchange, as well as cause infections. Unlike intact bacteria, OMVs are non-replicating but immunogenic, making them a safer option for use and direct administration [79].

The structural components of the OMVs are similar to the cell membrane and include nucleic acids, lipids (phosphatidylglycerol and phosphatidylethanolamine), proteins (membrane proteins: OmpA, OmpC, and OmpF; periplasmic proteins: AcrA), virulence factors, adhesion proteins, and proteins needed for host cell invasion [80,81]. The bacterial membrane, as well as OMVs, both show the presence of antigens specific to their origin and various pathogen-associated molecular patterns (PAMPs), which stimulate the immune system through uptake by DCs and activation of T cells [82]. The presence of a wide variety of PAMPs, higher immunogenicity, efficient lymphatic drainage, enhanced tumor retention, activation of immune cells at the tumor site, ease of production, and the flexibility to genetically modify the structure have attracted researchers’ attention toward the use of bacterial membranes and OMVs as cancer vaccines. The OMVs derived from Salmonella exerted antitumor activity by increasing the production of IFN-γ, IL-12, and TNF-α [83]. To enhance the therapeutic potential of cancer therapy, various chemotherapeutic, radioactive, and photoactive molecules can be attached physically or chemically to OMVs for a robust antitumor treatment regimen [84,85].

Recently, it has been reported that gram-positive bacteria also release a vesicle-like structure called membrane vesicles (MVs). The MV formation is quite challenging due to the presence of a thick peptidoglycan wall. A study conducted to examine the vesicle formation of S. aureus concluded that cytoplasmic vesicle formation occurs in response to elevated levels of phenol-soluble modulins, which alter membrane fluidity [86]. Unlike gram-negative bacteria, not all gram-positive bacteria secrete membrane vesicles. The components of MVs include lipopolysaccharides, nucleic acids, and proteins. The lipid profile of the MVs is not similar to that of its cell of origin and has different aggregation patterns. These MVs are involved in cellular processes such as biofilm formation, immune regulation, stress response, and communication. The bacterial membrane and OMVs’ coating help reduce off-target therapeutic effects, improve cell specificity, and increase cellular uptake. The fundamental concept of utilizing bacteria to enhance antitumor immunity has been established for over a century [87]. However, ongoing research in this field needs to focus on addressing challenges such as minimizing batch-to-batch variation, the presence of immunogens, and specific cell targeting approaches [88].

**Table 1 pharmaceutics-15-01677-t001:** Overview of membrane sources and their attributes.

Membrane Source	Surface Markers/ Proteins	Key Advantages	Limitations	Ref.
Erythrocytes	CD47, C8 binding protein	Prolonged circulation time; ease of isolation; reduced susceptibility to macrophage uptake	Absence of tumor-specific ligands	[20]
Platelets	CD62p, PECAM-1, CD44, CD47	Prolonged circulation time; ease of isolation; robust immune evasion	Aggregation of coated nanoparticles; absence of tumor-specific ligands	[26]
Macrophages	CCR2, VCAM-1, ICAM-1	Facilitates immune cell trafficking towards tumor; evades reticuloendothelial system; trans-endothelial migration through intercellular adhesion	Limited tumor-targeting ability	[32]
Dendritic cells	Peptide/MHC Complex, ICAM-3, CD40, CD44, CCR7	Upregulated co-stimulatory molecules; antigen-specific T-cell activation	Poor sensitivity and specificity of peptide/MHC complex to bind to CD8^+^ cells	[89]
Natural killer cells	NKG2D, NKp44, NKp46, NKp30, DNAM-1	Tumor recognition, a wide range of tumor targeting	Restricted proliferation of primary NK cells; lower infiltration in solid tumors	[90]
Cancer cells	CCAM, CD44, IG-SF	Strong adhesion among homotypic tumor cells, source of tumor antigens	Laborious isolation process; cell culture conditions, passage number, and genetic drift can induce variability in surface markers	[42]
Mesenchymal stem cells	CXCR4, PDGFR, VEGFR, E-selectin, P-selectin	Natural affinity toward tumor cells	Varying composition of the cell membrane may lead to ineffective therapeutic response	[48]
Exosomes	CD9, CD63, Alix, EP-CAM, CD55, CD59, MHC	Low immunogenicity; efficient cellular uptake; intrinsic tumor targeting	Laborious isolation process; presence of inherent biological cargo can cause unwanted biological effects	[57]
Viral Capsids	AAV-Rep78, Parvovirus NS1	Activates host immune system; selective apoptosis of tumor cells	Non-specific binding to healthy cells may lead to immunogenic responses	[65]
Bacteria	OmpA, OmpC, AcrA	Immune cell activation; self-adjuvant characteristic	Pathogenicity needs to be adequately addressed before in vivo use	[82]

## 3. Preparation of Membrane-Coated Nanosystems

A schematic representation of all processes involved in preparing a membrane-coated nanosystem, starting from membrane isolation, selection of the NP core (along with its loaded cargo), and membrane coating, is shown in Figure 2.

### 3.1. Membrane Isolation/Extraction

To fabricate membrane-coated nanosystems, the first step is to obtain a functional cell membrane from the desired source. Commercially available infrastructures for blood collection and processing make it easy to acquire membranes from primary blood cells such as erythrocytes, platelets, and immune cells. In contrast, cell lines or bacterial strains can be cultured in a laboratory setting to obtain sufficient membranes for preclinical studies. Suspension cells can be grown volumetrically in shaker or spinner flasks, which simplifies the harvesting process compared to adherent cells that require enzymatic or physical detachment. Once enough source cells have been obtained, the membrane material is derived. For anucleate cells, hypotonic treatment or freeze-thaw cycles can be used to release intracellular contents, followed by high-speed centrifugation to form a pellet containing the membrane [91]. However, the process is more complex for nucleated cells, which require the separation of the plasma membrane from intracellular organelles and proteins. Mechanical homogenization, sonication, or nitrogen cavitation can be used for cell lysis, followed by differential or gradient centrifugation to isolate the plasma membrane [92]. In recent times, there has been a surge in the utilization of different commercial kits for membrane isolation, as observed in the literature [93]. Nonetheless, the high cost of these kits restricts their scalability. A comparable challenge is encountered in the isolation of extracellular vesicles, although the dependence on isolation kits has been reduced due to advancements in hollow fiber bioreactors [94].

### 3.2. Selection of NP Core

Depending on the intended application, choosing the appropriate core is critical in determining the type of cargo that can be loaded into it. In the field of cancer theranostics, the cargo could include drug molecules, biomolecules such as siRNA, miRNA, and mRNA, immune adjuvants, or contrast agents. A range of NP core designs can be used to create membrane-coated nanosystems, with materials ranging from natural biomaterials such as chitosan, alginate, and silk fibroin to complex synthetic systems such as polymeric, lipid-based, or metallic NPs [95,96]. Before in vivo application, the toxicological aspects of nanomaterials must be given significant consideration. The membrane coating serves as an intermediate barrier between the core and the biological system, providing additional benefits to the core, such as increased biocompatibility [97]. The physicochemical properties of the core NPs, such as size, shape, charge, and elasticity, play a vital role in the efficiency of cell membrane coating and drug loading [98]. Positively charged NPs tend to collapse the cell membrane arrangement, leading to the formation of particle aggregates during coating. Negatively charged NPs are better suited for successful cell membrane coating due to their strong electrostatic interaction with the cell membrane [99]. Additionally, it is generally observed that denser (non-hollow) cores facilitate more efficient membrane coating. Attributes such as core size, curvature, and chemical composition impact the surface area, size-to-volume ratio, elasticity, and mechanical properties, which, in turn, influence cellular uptake, bio-nano interactions, and the integrity of the cell membrane coating [100]. Different NP cores functionalized via cell membrane coating are listed in Table 2.

### 3.3. Membrane Coating

Cell membrane-coated NPs can be fabricated using several different methods, such as physical extrusion, sonication, and microfluidic coating. These techniques utilize the electrostatic interactions between the NP core and membrane components to create a core–shell structure that is stable and has the right-side-out membrane topological orientation, which is energetically beneficial [123]. Initially, physical extrusion was the only method used, where a porous membrane was used to co-extrude nanoparticulate cores and purified membranes. The process of physical extrusion was adapted from the production of synthetic liposomes, and it is believed that the mechanical force applied during extrusion disrupts the membrane structure, allowing it to reform around the nanoparticulate cores [124]. A sonication-based approach has been implemented to fabricate core–shell nanostructures, and this technique involves subjecting the two components to ultrasonic energy, causing disruptive forces that lead to the spontaneous formation of the desired nanostructure. The process of membrane coating is thought to occur through a complex interplay between the inherent characteristics of the bare nanoparticulate cores and cell membrane-derived vesicles. It is believed that the semi-stable nature of the nanoparticulate cores, combined with the asymmetric charge distribution of biological membranes, creates an energetically favorable environment for the formation of a core–shell configuration with a right-side-out membrane orientation [125]. Overall, both extrusion and sonication techniques utilize physical forces—shear force and acoustic force, respectively—to either deform or rupture membrane vesicles, enabling them to reassemble around nanoparticulate cores in an energetically favorable process [126]. There are also innovative techniques, such as microfluidics, that have been developed for the encapsulation of NPs within cell membranes, offering tremendous potential for a wide range of applications.

#### 3.3.1. Physical Extrusion

Co-extrusion of nanoparticulate cores and purified membrane vesicles through a porous membrane is a widely recognized method for physically coating cell membranes onto NPs [29]. Extrusion offers several advantages, including a high degree of precision and reproducibility facilitated by syringe-based commercial extruder devices. The resulting samples are free from contaminants larger than the defined pore size and exhibit a monodisperse size distribution, ensuring consistent batch-to-batch reproducibility [126]. The physical extrusion approach has been widely utilized in numerous studies involving the coating of nanoparticulate systems with cellular membranes. For instance, Cao et al. demonstrated the potential of a targeted drug delivery system for the treatment of pancreatic carcinoma by utilizing neutrophil membrane-coated PEG-PLGA NPs loaded with celastrol (Figure 3A). Through a process of co-extrusion, the neutrophil plasma membrane vesicles were coated onto the PEG-PLGA cores by passing them repeatedly through a 220 nm polycarbonate membrane. The process facilitated efficient coating of the neutrophil plasma membrane onto the PEG-PLGA nanoparticulate system, and the resulting neutrophil membrane-coated NPs (NNPs) exhibited an average size of 167.4 ± 2.6 nm and a PDI of 0.215 ± 0.037. Upon negative staining and TEM visualization, the NNPs obtained were found to have a characteristic core–shell structure with a single dimmer layer (Figure 3A). The thickness of the plasma membrane layer was observed to be in the range of 10–20 nm. It has been suggested that the membrane coating process involves the translocation of the plasma membrane bilayer onto the NP surface, resulting in a right-side-out conformation that preserves the membrane’s functionality [127]. Likewise, biomimetic PLGA NPs coated with human cancer cell membrane fractions (CCMFs) have been developed, wherein physical extrusion allowed the PLGA NPs to be coated with a ~5 nm thick plasma membrane, resulting in the formation of CCMF-PLGA NPs [128].

By coating biodegradable polymeric NPs with natural erythrocyte membranes, including both membrane lipids and related membrane proteins for long-circulating cargo transport, Hu et al. reported on a top–down biomimetic strategy in particle functionalization. The RBC-membrane-derived vesicles were fused with the PLGA NPs (70 nm) by subjecting them to physical extrusion several times through a 100-nm polycarbonate porous membrane using an Avanti mini extruder. The use of mechanical force during the physical extrusion process allowed for the fusion of sub-100-nm PLGA NPs with the lipid bilayers, resulting in vesicle-particle fusion [22]. Top–down assembly for developing biomimetic drug delivery platforms has also been carried out by Xuan et al. Mesoporous silica nanocapsules (MSNCs) were mixed with freshly made macrophage cell membrane (MPCM) vesicles and then extruded (20 times) over a 100 nm porous polycarbonate membrane to produce MPCM-camouflaged MSNCs (Figure 3B). These MSNCs were subsequently loaded with DOX and used as cancer treatments [129].

Furthermore, research has been conducted to better understand the dynamics of natural cellular membranes and polymeric nanoscale substrates in relation to cellular membrane-cloaked nanodevices. For instance, Luk et al. studied the interfacial aspects of the red blood cell membrane-cloaked NPs (RBC-NPs). They investigate the impacts of polymeric particles’ surface charge, completeness of RBC membrane coverage, sidedness of RBC upon coating, and surface curvature on the membrane cloaking process. To facilitate the fusion of RBC membrane vesicles with the PLGA cores, an Avanti mini extruder was used to physically extrude a mixture of PLGA particles and RBC membranes through a 100-nm polycarbonate porous membrane. The study demonstrated that negatively charged polymeric NPs could be effectively coated with RBC membranes in a right-side-out manner, likely due to the repulsive forces between the particle and extracellular membrane surfaces. In contrast, positively charged polymeric cores showed rapid aggregation upon mixing with RBC membranes due to strong electrostatic attractions causing membrane-particle bridging. The results indicate that the RBC membranes can provide complete coverage of the negatively charged polymeric NPs, leading to their enhanced colloidal stability. Moreover, substrate properties and surface glycans on RBC membranes also influence the assembly of membrane–particle complexes [130]. Biomimetic NPs with improved characteristics for tumor diagnostics and drug delivery have been developed using the physical extrusion process.

#### 3.3.2. Sonication

Sonication-based approaches are another choice, where a disruptive force provided by ultrasonic energy leads to the spontaneous fusion of NPs with membrane vesicles to form a core–shell nanostructure [131]. Since sonication makes use of acoustic energy, the interactions between NPs and membrane vesicles in co-suspension can be affected by a number of factors. Besides the commonly reported sonication parameters such as power, amplitude, and time, other factors such as the volume of the suspension, the temperature of the bath, the density of the particles and membrane vesicles, and the volume of the suspension also impact the formation of bio-inspired nanosystems. Moreover, sonication results in higher formulation yield and no material loss as syringe dead volume [126].

Using sonication, Kang et al. developed carfilzomib (CFZ)-loaded PLGA NPs coated with neutrophil membranes as a nanosize neutrophil-mimicking drug delivery system (NM-NP) to target CTCs and prevent the creation of a metastatic microenvironment. The essential processes that enable the creation of the structure of polymeric NPs with cell membrane cloaking were thought to be hydrophobic interaction and electrostatic repulsion between the negatively charged PLGA cores and the cellular membrane. A single dimmer neutrophil membrane layer (10 nm thick) was detected by TEM on the surface of NM-NP-CFZ (Figure 4A). The neutrophil membrane had relatively good coating effectiveness on PLGA NPs, with a membrane-to-polymer ratio of 1:1. The membrane coating enhanced the stability of NPs and prevented their aggregation [132]. Biomimetic targeting of antigen-specific immune cell populations for preventing immune cell malignancies has also been investigated utilizing NPs functionalized with natural membranes derived from cells expressing the cognate antigen. Using RBC-specific B cells as a model target, Luk et al. demonstrated that RBC membrane-coated NPs (Figure 4B) exhibit increased affinity compared with control NPs. Sonication was used to cause membrane fusion between the PLGA cores and the RBC membranes in order to create RBC-NPs. The RBC-NPs were around 90 nm in size, and the polymeric cores were seen to have a core–shell structure, indicating the presence of a membrane coating (Figure 4B) [133].

#### 3.3.3. Microfluidic Coating

Microfluidic devices permit streamlined mixing of membrane vesicles and NP cores; nonetheless, external force in the form of sonication or electroporation is necessary for the complete coating of NP cores [126]. Recently, Liu et al. illustrated the promise of using a microfluidic sonication approach to fabricate biomimetic NPs for improved biocompatibility and targeting efficacy. Using a microfluidic sonication method, PLGA NPs coated with either exosome membrane (EM) or cancer cell membrane (CCM) and encapsulated with imaging agents were created in a single step for immune evasion-mediated tumor targeting. The combined effects of acoustic pulses and hydrodynamic mixing were employed in the one-step microfluidic synthesis of EM-, CCM-, and lipid-coated PLGA NPs. Microfluidic sonication is the process of immersing a two-stage microfluidic device in an ultrasonic bath before initiating the production of membrane-coated PLGA NPs. The microfluidic device is composed of one straight channel and one spiral channel that are coupled to four inlets (Inlets 1–4) and one outlet (Outlet 5) (Figure 5A). EM (or CCM) in PBS, PLGA in organic solution, and PBS solutions are injected into Inlets 1, 3, and 4, respectively, to prepare EM- or CCM-coated PLGA NPs. To prepare lipid-PLGA NPs, DI water, PLGA in organic solution, and lipid in PBS are injected into the device through Inlets 1 and 3, Inlet 2, and Inlet 4, respectively. After placing the apparatus in an ultrasonic bath, the EM-PLGA NPs, CCM-PLGA NPs, and lipid-PLGA NPs that are produced can be extracted through Outlet 5. High flow velocities (>3 m/s) result in improved hydrodynamic mixing, which in turn aids in the interfacial nanoprecipitation of uniform PLGA cores inside the microchannel. In this study, microfluidic sonication created intense compressive pressure, which allowed for almost simultaneous (<30 ms) coating of a variety of membranes onto the PLGA cores with remarkable coating efficiencies (up to 93%). Similar core–shell structures and hydrodynamic diameters (200 nm) with excellent monodispersity were observed in the resultant EM-PLGA NPs, CCM-PLGA NPs, and lipid-PLGA NPs (Figure 5A). Microfluidic sonication offers distinct advantages over conventional approaches for producing biomimetic NPs with similar sizes and core–shell structures, including improved coating of a variety of biomembranes, rapid formation of core–shell NPs, integration of multiple reactions onto a single chip, and high encapsulation efficiency of active compounds [134].

Similarly, a microfluidic device combining rapid mixing and electroporation was effectively used to coat the RBC membrane around magnetic NPs for improved imaging-guided cancer treatment. Two inputs, a Y-shaped merging channel, an S-shaped mixing channel, and an electroporation zone before the outlet made up the five main components of the device. RBC-derived vesicles were entirely fused with the magnetic NPs (MNs) and subsequently flowed through the electroporation zone. The electric pulses between two electrodes could effectively promote both the introduction of MNs into RBC-vesicles and the subsequent facilitation of the production of RBC-MNs (Figure 5B). High-quality particles with complete coatings and outstanding stability were created by carefully adjusting the pulse voltage and duration as well as the flow velocity [135]. In general, the process pulse voltage, duration, and flow velocity must all be tuned for this method to be effective [123].

### 3.4. Other Special Approaches

In addition to methods that employ isolated membrane components, there is a novel strategy based on the in vivo packaging/covering of nanomaterials with living cells [125,131]. By initially incubating cells with iron-oxide NPs, gold NPs, or quantum dots, Andriola et al. developed a fabrication scheme. The cells’ ability to secrete vesicles containing the foreign NPs was subsequently demonstrated by incubating them in serum-free media [136]. Despite developments in other methods, extrusion and sonication continue to be the most widely used for membrane coating on NPs [126].

## 4. Methods of Characterization

### 4.1. Physicochemical Properties

#### 4.1.1. Size and Surface Attributes

Real-time evaluation of the membrane-coated nanosystem can be performed using dynamic light scattering (DLS). This technique measures the hydrodynamic diameter of the NPs and can determine the size difference between the particles pre- and post-coating, providing insights into the formation of the cell membrane coat on the NPs. The thickness of the cell membrane coating depends on the number of layers that are coated and the degree of fusion between the cell membrane and NPs. A zeta sizer can be utilized to evaluate the surface charge of the particles. The zeta potential provides information about their stability and interactions with biological systems. If the cell membrane coating is uniformly performed, a shift in the zeta potential towards a negative value (charge of the cell membrane) is expected. The colloidal stability of the membrane-coated nanosystem upon storage for a long time can also be evaluated by checking these properties [137,138]. Figure 6A shows the graph of hydrodynamic diameter and surface charge of mesoporous silica NPs before and after modifying them with the macrophage cell membrane. The size increase from 47.8 nm to 65.1 nm indicates the coating of the macrophage cell membrane around the mesoporous silica NPs. Similarly, the value of the surface charge is changed from −7.5 mV to −16.9 mV, which is considered an indication of coating [129]. This technique serves as a preliminary test to confirm the extent of coating and ensure the uniformity of the coating on the NPs.

Membrane-coated nanosystems can be visualized using microscopic techniques such as transmission electron microscopy (TEM), scanning electron microscopy (SEM), and fluorescence microscopy [139]. Through TEM imaging, a visual representation of the core–shell structure of the coated NPs is obtained. Figure 6A depicts the core–shell structure of the silica NPs coated with the macrophage cell membrane. The core appears as a denser region because of the presence of gold NPs in the core, while the shell appears as a thinner layer surrounding the silica NP. This contrast in density can be attributed to the difference in electron densities between the NP and the cell membrane. The hollow cell membrane particle appears as a white circle. The change in size at the pre- and post-coating stages gives the thickness of the membrane coated, which is approximately 10–20 nm. Before imaging, for better resolution as well as to visualize the cell membrane, the nanosystem is stained with negative stains (such as phosphotungstic acid or uranyl acetate) [140,141].

Another technique used in imaging is confocal laser scanning microscopy (CLSM). The cell membrane and NPs are tagged with different fluorescence dyes before coating. The fluorescence images of the NPs give an estimate of how the particles are coated and the integrity of the cell membrane after coating [22,142]. UV–visible spectroscopy can also be utilized as a qualitative method to verify the presence of a cell membrane coating on NPs. Figure 6B shows the UV–visible spectra of the uncoated magnetic clusters (MNC), the RBC membrane, and the RBC membrane-coated NPs (MNC@RBC). The appearance of a new peak in the spectrum, in addition to the peak of uncoated NPs, would indicate the coating of a cell membrane around the NPs [143,144].

#### 4.1.2. Membrane Properties

Membrane integrity, permeability, and coverage can be evaluated to characterize a membrane-coated nanosystem. To evaluate the integrity of the cell membrane coating on NPs, a commonly employed technique is the fluorescence quenching assay [145]. The NPs are initially labeled with a fluorescence dye, nitro-2,1,3-benzoxadiazol-4-yl (NBD), and later coated with the cell membrane. In the presence of a negatively charged reducing agent, dithionite, NBD is reduced irreversibly to amino-2,1,3-benzoxadiazol (ABD), which is a non-fluorescent molecule. Dithionite (DT) is non-permeable to the cell membrane, and hence, if the particle is fully coated, there will be no reduction in fluorescence after adding DT to the coated NPs. Figure 6C shows the CLSM images of membrane-coated SiO_2_ NPs, giant unilamellar vesicles (GUVs), and endocytosed SiO_2_ NPs (E-SiO_2_) before and after the addition of the DT. Higher fluorescence intensity would indicate successful membrane integration, suggesting that the NPs are fully coated with the cell membrane. On the other hand, lower fluorescence intensity would suggest partial or no coating of the NPs with the cell membrane. It enables precise assessment of the integrity of the coated cell membranes, providing valuable insights into the effectiveness of the coating procedure and the functional properties of the resulting cell membrane-coated NPs [99]. The information obtained from this assay can be utilized to optimize strategies for the fabrication of functionalized NPs with enhanced performance in various biomedical applications, such as drug delivery, imaging, and theranostics.

The inherent property of selective permeability of the cell membrane enables the use of membrane-coated nanosystems as carriers for delivering drugs, nanoreactors for delivering enzymes, and templates for nanomaterial synthesis. After coating the NPs with the cell membrane, the permeability should be retained to release the encapsulated drug. This can be evaluated by using a fluorescence probe with membrane permeability. The presence of a probe inside the nanosystem, when imaged using CLSM, confirms the coating and functioning of the cell membrane [146]. The extent of membrane coverage in the NP system was first calculated by using an aggregation assay. This assay is based on the conjugation of streptavidin with biotin and the formation of a stable aggregate. Luk et al. prepared biotin-loaded PLGA NPs, which were later coated by RBC-derived cell membrane (size~100 nm). After coating the PLGA NPs with different amounts of RBC-derived cell membrane via the extrusion method, the resulting NPs were mixed with the solution of streptavidin. If the coating is uniformly formed, the biotin molecule will not react with the streptavidin, and no change in the size of coated NPs is observed. If the polymeric core is not fully coated (membrane-to-polymer ratio less than 25 µL mg^−1^), aggregate formation occurs (size~2000 nm). This study reported that 85 µL of the membrane is required to fully cover a milligram of polymeric core NPs [130]. 

Another technique to study the extent of fusion as well as calculate the percentage of coverage of the cell membrane around the NPs is fluorescence resonance energy transfer (FRET). This technique calculates the molecular distance between the fluorophores from the pair of donor and acceptor dyes (such as NBD and Rhodamine, DiL, and DiO). The cell membrane is labeled with one of the dyes from the pair, and the NP core is loaded with the other. From the donor molecule, energy transfers to the acceptor molecule, which is measured to obtain the fusion as well as the coating of the membrane onto the NPs. Figure 6D compares the fluorescence intensity of the MCF-7 membrane-coated nanosystem (CCM@LM) with that of PEGylated liposomal yolk (LM). The fluorescence intensity of CCM@LM increased rapidly within 60 min, which confirms the cell membrane fusion with the NP [147].

**Figure 6 pharmaceutics-15-01677-f006:**
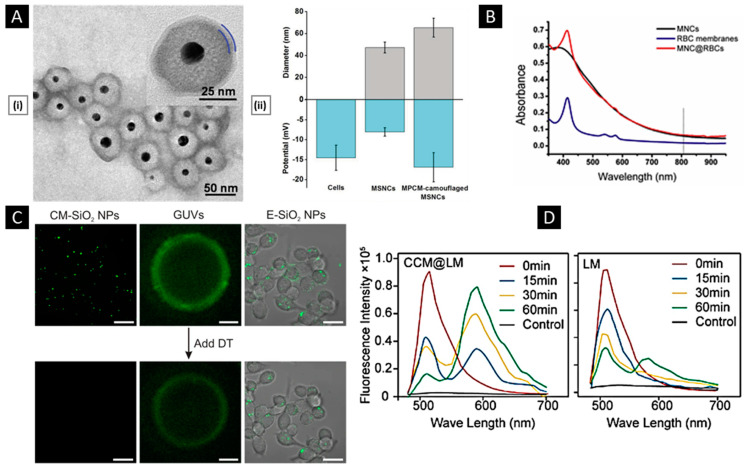
(**A**) (**i**) TEM images of a macrophage membrane coated with a silica core negatively stained with uranyl acetate depict the successful shell-like assembly of a membrane with a thickness of about 6 nm. (**ii**) Representative hydrodynamic diameters (gray) and surface zeta potentials (blue) of the macrophage cells, isolated membrane vesicles, and final membrane-coated nanosystem. Reproduced with permission from [129], copyright Wiley-VCH 2015. (**B**) UV–vis spectra of MNCs and MNC@RBCs. Reproduced with permission from [143], copyright Elsevier 2016. (**C**) CLSM images of membrane-coated SiO_2_ NPs, GUVs, and E-SiO_2_ before and after the addition of the DT (Scale bar: 20 μm). Reproduced with permission from [99], copyright Springer Nature 2021. (**D**) Fluorescence intensity comparison of CCM@LM with LM Reproduced with permission from [147], copyright American Chemical Society 2019.

### 4.2. Biological Properties 

#### 4.2.1. Verifications of Membrane Proteins

The physicochemical characterizations mentioned above offer significant insights into the level of effective cell membrane coating. However, a more crucial aspect is the evaluation of membrane proteins in their functional state in the final nanosystem. This assessment is essential, as the presence of these proteins is the main driving force behind the ultimate biomedical applications.

The verification of membrane proteins involves the identification, separation, and refining of proteins, along with an assessment of their structural and functional characteristics. Several methods have been suggested for accomplishing these objectives. Amino acid analysis is a quantitative technique for determining protein concentration, amino acid composition, and protein and peptide content [148]. Another method is the use of UV-vis spectroscopy to quantitatively detect functional groups and identify materials by comparing their absorbance patterns. A more precise method of verifying the intact molecular weight of membrane proteins is through the use of mass spectrometry (MS). This technique operates by measuring the mass-to-charge ratio of proteins present in a solution. Additionally, by coupling N-terminal Edman chemistry with MS, it is possible to determine the amino acid sequence of an entire protein [149]. Peptide mapping via chemical or enzymatic digestion and TANDEM MS can also be employed to further characterize proteins [150]. While numerous sophisticated characterization methods are available, sodium dodecyl sulfate-polyacrylamide gel electrophoresis (SDS-PAGE) and Western blotting analysis can suffice to verify membrane proteins for most purposes.

SDS-PAGE is a widely used laboratory method that allows the separation and analysis of proteins based on their size. The technique involves the use of a gel matrix consisting of polyacrylamide, along with a detergent (SDS) and a reducing agent (such as β-mercaptoethanol or dithiothreitol) that denatures and imparts a uniform negative charge to the proteins. The protein mixture is then applied to the gel, and an electric field is applied, causing the proteins to migrate through the gel matrix according to their molecular weight. Smaller proteins move faster than larger ones, resulting in the separation of proteins based on their size. Staining techniques, such as Coomassie blue, can be used to visualize the separated proteins. SDS-PAGE is commonly used to compare the protein profiles of source cell membranes, extracted membranes, and derived NPs [151]. This technique can indirectly assess the efficiency of the membrane extraction/purification method and the suitability of the membrane coating technique. Comparable protein profiles and the preservation of protein markers are considered to be good indicators of retained biological functions after membrane coating [152]. To confirm the presence of specific protein markers, the separated proteins are transferred onto a membrane (such as nitrocellulose or polyvinylidene difluoride), and Western blotting is performed using highly selective antibodies (Figure 7A) [153].

#### 4.2.2. In Vitro Functional Validation

To examine the biological activity of membrane proteins expressed on a nanosystem’s surface, it is possible to analyze their effects at the cellular level before testing them on higher animal models. For nanosystems created using blood-sourced cells (such as erythrocytes or immune cells), the system should have an extended systemic circulatory time. This feature depends on the cell’s inherent ability to avoid macrophage phagocytosis. An indirect assessment of this property can be performed in vitro through a comparative uptake assay. A recent study conducted by Liu et al. utilized RAW264.7 macrophage cell lines to assess the impact of membrane coating on the uptake. The cells were cultured with Rhodamine B-labeled nanosystems (with an uncoated core as a control group) for 60 min, followed by visualization through confocal microscopy. The results showed a significantly lower mean fluorescence intensity in the membrane-coated system (Figure 7B) [154]. On a similar note, efficient and selective homotypic targeting is a key goal when using cancer cell membrane-coated nanosystems. Lehto et al. conducted a study where they coated SiO_2_ NPs with CT26 cell membrane, resulting in a system known as CM-SiO_2_. To assess the homotypic targeting capacity of CM-SiO_2_ in vitro, the authors used CT26 cells (murine colorectal carcinoma), HeLa (cervical cancer cell line), and MCF-7 (breast cancer cell line) to conduct a comparative uptake assay. All cell lines were cultured for 24 h and then exposed to the tested samples (50 μg/mL) in a fresh medium. Particle uptake was assessed post-incubation and processing using CLSM, revealing that CM-SiO_2_ demonstrated preferential uptake in source cells. Additionally, the authors improved the membrane coating integrity by modifying CT26 membranes with dipalmitoylphosphatidylcholine (DOPC) liposomes (HM-SiO_2_), resulting in a more selective and profound homotypic uptake (Figure 7C) [155].

## 5. Functionalization Approaches

As research in cell membrane-coated nanosystems progresses toward their utilization in complex biological systems, the need for their multifunctionality also increases. By incorporating additional functions (beyond those derived from the natural membrane source), applications can be drastically broadened [156,157]. To achieve this, conjugation methods that employ amine-, carboxyl-, biotin-, or sulfhydryl-based reactions were initially explored [158]. While convenient, these methods lack control over the position/density of the linked ligands. Such random chemical alteration can compromise membrane integrity by inducing membrane protein aggregation or trigger unwanted immune effects by exposing phosphatidylserine from membrane bilayers [159]. To counter these shortcomings, a handful of nondisruptive and straightforward functionalizing strategies that preserve the natural function of source cells have been reported [160]. Figure 8 provides a schematic overview of these emerging functionalization approaches, and the ensuing section discusses some recent examples. 

### 5.1. Lipid Insertion

This technique involves the use of ligands of interest conjugated to a lipid anchor, which is mixed with a source membrane. This process exploits the fluidity of bi-layered lipid membranes rather than a chemical process, making it an effective physical insertion approach [161]. Ease of incorporation (via sonication or extrusion) and precise control over ligand density (through controlled initial input) make lipid insertion an appealing functionalization method. DSPE-PEG (i.e., 1,2-distearoyl-sn-glycero-3-phosphoethanolamine-N-[amino (polyethylene glycol)-2000]) and its derivatives are the preferred lipid anchor for inserting small molecule-based affinity ligands such as folate [162], mannose [163], Cyclic Arg-Gly-Asp [164], and several targeting aptamers into source cells (mostly of non-cancerous origin). In a study by Fang et al., receptor-specific targeting ability was incorporated into a long-circulating erythrocyte membrane-coated nanosystem via lipid insertion of the nucleolin-targeting aptamer AS1411, which contained a 3′ thiol modifier that was conjugated to lipid-PEG-maleimide (via maleimide–sulfhydryl chemistry). The conjugate was then incubated with the emptied RBC membrane and extruded with the core NP to assist membrane incorporation. The modified membrane allowed for a 2-fold increase in uptake of the erythrocyte membrane-coated system in cancer cells (vs. a non-functionalized control), quantified using fluorescence microscopy (Figure 9A) [165].

Apart from small molecules, antibodies can also be included in lipid membranes. To achieve this, functional groups that are reactive to antibodies are linked to lipid molecules first and then used for insertion. Cancer-specific surface markers such as human epidermal growth factor receptor 2, epithelial growth factor receptor, and epithelial cell adhesion molecule have been the focus of exploration [166,167]. However, due to antibodies’ bulkiness, it is difficult to maintain their functional geometric orientation on the membrane surface during particle coating. To overcome this challenge, smaller antibody fragments can be used to enhance spatially selective conjugation with lipid anchors [168]. In addition to their supporting role as anchors for ligand attachment, lipids can serve functional roles by providing responsive properties to stimuli such as light, hypoxia, or pH. For example, Su et al. incorporated DiR (1,1′-dioctadecyl-3,3,3′,3′-tetramethylindotricarbocyanine iodide) into the erythrocyte membrane to convert NIR light radiation into heat and induce local hyperthermia. The system’s core was prepared using a thermosensitive lipid with a transition temperature of around 41.5 °C. Under NIR laser stimuli, DiR embedded in the membrane produced thermal energy to trigger the phase transition-mediated degradation of the NP core, resulting in a burst release of an entrapped anticancer drug [169]. In a related study, Liu et al. incorporated a pH-sensitive lipid (DSPE-polyethyloxazoline) with a PLT membrane via co-extrusion. When exposed to acidic endosomal pH, the polyethyloxazoline moiety underwent rapid protonation, generating electrostatic repulsion to destabilize the membrane structure and release the therapeutic payload [170].

### 5.2. Hybridization

It refers to the process of fusing cell membranes that are procured from two distinct source cells, intending to endow the resultant “hybrid” membrane with key proteins and specific biomimetic properties of the parent cells that can facilitate multifunctional applications [171]. The simplest way to generate hybrid membranes is the isolation of membranes from individual cell types (the ratio of which depends on the desired function), followed by their mechanical fusion via extrusion or sonication [172]. Alternately, they can also be generated by first fusing different live cells, followed by deriving the membrane from the cell hybrids (Figure 9B) [173]. The main goal of the hybridization approach is to address any limitations or drawbacks of individual membrane-based systems or to incorporate additional features into the final nanosystem to fulfill critical functions. Membrane hybridization is primarily used to enhance the targeting ability of final membrane-coated nanosystems by introducing affinity ligands unique to one cell type to another. Dehaini et al. were the first to describe a hybridized cell membrane-coated nanosystem, which was created by fusing membranes from erythrocytes and PLTs. The resulting platform, termed [RBC-P]-NPs, inherited features from both cell types. It demonstrated an extended in vivo circulation and maintained functional surface markers on PLTs, which are crucial for improved site-specificity [174]. In a similar study, Wang et al. combined membrane materials derived from erythrocytes and melanoma cells. The resulting hybrid facilitated a prolonged circulation half-life (~10 h) along with highly specific self-recognition of the source cell when administered to tumor-bearing mice [142].

Another unique application of membrane hybridization is in cancer immunotherapy, where an “immune-stimulatory” attribute can be incorporated in addition to cancer targeting. Liu et al. reported a hybrid membrane-based immunotherapeutic platform developed by combining cancer and DC membranes. Through cellular fusion, this platform triggers immune activation and effectively expresses whole cancer antigens, which aids in self-targeting. Additionally, the platform incorporates immunological co-stimulatory molecules from DCs, resulting in a more robust form of immunotherapy [175]. Beyond mammalian cells, membranes from gram-negative bacteria and their OMVs are highly efficient in activating immunotherapy for cancer treatment. When fused with cancer cells, they localize within tumors and trigger immune responses by stimulating the production of antitumor cytokines. Wang et al. combined OMV derived from E. coli DH5α and B16-F10 cancer cell membranes to coat hollow polydopamine NPs. Upon intravenous administration via the tail vein in the melanoma-bearing mice, the system displayed selective accumulation in the melanoma area and upregulated the tumoral expression of IL-12p40 and IFN-γ [85]. In a similar study, Chen et al. developed a tumor-targeted cancer nano-vaccine by combining cancer cell membrane with OMV obtained from attenuated Salmonella. This system incorporated targeting antigens and natural immunostimulatory adjuvants, and during prophylactic testing, it successfully fought tumorigenesis by inducing a strong antitumor immune response [176].

### 5.3. Metabolic Engineering

Metabolic engineering involves the manipulation of cells’ natural biosynthetic pathways in order to regulate cellular properties. In the case of cell membrane modification, metabolic substrates are combined with functional moieties and then introduced to cells for uptake and metabolism. By employing this technique, non-natural conjugates can take over natural biosynthesis pathways, integrate with relevant cellular metabolic processes, and eventually attach to cell surfaces. From this fundamental concept, two approaches, namely glycoengineering and lipid engineering, have been developed.

Glycoengineering involves the production of oligosaccharides and glycoconjugates, which can be used to modify cell membranes. This process relies on several pathways, such as fucose salvage, sialic acid salvage, and N-acetylgalactosamine (GalNAc) salvage, to produce specific monosaccharide substrates. These substrates, including N-acetylmannosamine (ManNAc), N-acetylneuraminic acid (Neu5Ac), GalNAc, and fucose, can then be combined with functional groups to form conjugates that can modify the functional characteristics of the cell membrane. Han et al. recently conducted a study where they used tetra-acetylated N-azidoacetylgalactosamine (Ac4GalNAz) to introduce azide groups onto T cell membranes through the GalNAc salvage pathway. Following this, the N_3_-labeled T cell membranes were derived and coated onto photosensitizer-loaded PLGA cores (termed N_3_-TINPs). Additionally, a Bicyclo [6.1.0] nona-4-yne (BCN)-modified mannose substrate (Ac4ManN-BCN) was administered into the tumor region, where it was taken up by tumor cells via the sialic acid pathway and expressed on the tumor surfaces. After the injection of N_3_-TINPs, a selective click reaction occurred between the BCN and N_3_ groups, facilitating the specific homing of N_3_-TINPs to the tumor region. This tumor-specific homing was further aided by the immune recognition of CD3 on the T cell membrane of N_3_-TINPs by the tumor cells. As a result, the nanosystem accumulated at higher levels in the tumor region after intravenous administration compared to NPs coated with unmodified T-cell membranes (Figure 10A) [177].

Lipid engineering refers to the manipulation of cellular membranes through natural lipid synthesis pathways, such as the cytidine 5’-diphosphocholine (CDP-choline) pathway. This process employs choline analogs to introduce bioorthogonal linkers onto the cell membrane, enabling subsequent conjugation with functional ligands. Zhang et al. utilized an azide-choline substrate to introduce N_3_ groups onto the leukocyte membrane via the CDP-choline biosynthesis pathway. The resulting N_3_-labeled membrane was then coated onto magnetic nanoclusters (MNCs). By means of click chemistry, the N_3_-tagged MNCs were conjugated with major histocompatibility complex class I and the co-stimulatory ligand anti-CD28. In a murine lymphoma model, the nanoclusters acted as artificial antigen-presenting cells and effectively stimulated the proliferation of CD8^+^ T cells, surpassing that induced by free anti-CD28. When administered via a single intratumoral injection, the nanoclusters led to a gradual yet significant increase in the number of tumor-infiltrating lymphocytes per tumor, and tumor growth was significantly delayed within 14 days. No such effect was observed when free anti-CD28 was administered at an equal dose [178]. Employing the same phospholipid pathway, Zhang et al. prepared N_3_-labeled macrophage membranes that were further conjugated with an RGD peptide that targets integrin αvβ3 (overexpressed on tumors). The engineered membrane was coated onto the MNC–siRNA nanocomplex. When intravenously injected, it showed a 2.7-fold increase in the level of tumor accumulation (vs. the unmodified membrane system) [179]. Metabolic engineering can also be applied to modify bacterial membranes. For example, Price et al. reported the glycoengineering of OMVs from nonpathogenic bacteria to express the surface glycans of pathogenic strains. The immunogenic response of modified OMVs was assessed in a murine model to evaluate their potential as vaccine candidates. According to Western blotting, sequential administration of the modified OMVs resulted in the generation of high titers of pathogen-specific IgG antibodies in the sera of immunized mice. At day 42, the IgG response levels in mice injected with modified OMVs were comparable to those in mice injected with the commercially available Prevnar 13^®^ (a pneumococcal vaccine), indicating a similar level of immunogenicity. These findings suggest that the modified OMVs may confer a comparable level of protection. When combined with an appropriate adjuvant-loaded core and modified to possess a suitable tumor immunity-triggering antigen, this approach can be explored for immunization against cancer, offering promising prospects for cancer vaccine development [180].

**Figure 10 pharmaceutics-15-01677-f010:**
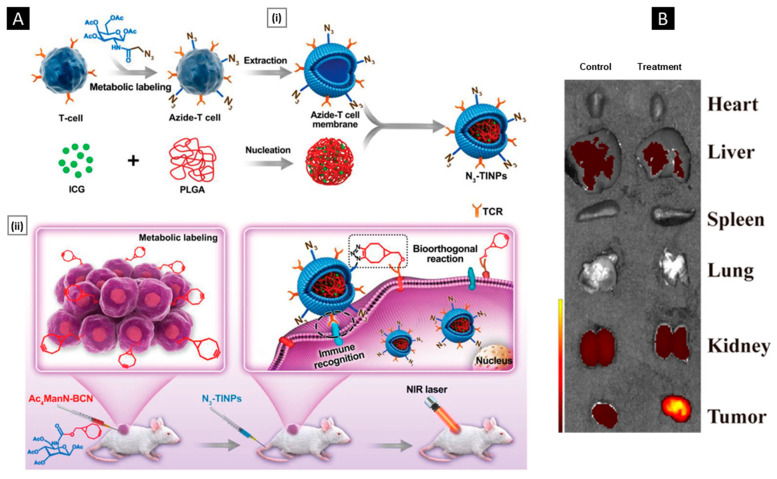
(**A**) (**i**) Schematic illustration depicting the synthesis of N_3_-TINPs. (**ii**) Modification of tumor cells to carry the BCN group via natural glycometabolic labeling by pretreatment with Ac4ManN-BCN and subsequent targeting with N3-TINPs for biomedical applications. Reproduced with permission from [177], copyright Wiley-VCH 2019. (**B**) Fluorescence images of organs from HepG2 tumor-bearing nude mice treated with preS1 ligand-expressing biomimetic oncolytic adenoviruses (left panel shows representative images from the control group). Reproduced with permission from [181], copyright American Chemical Society 2019.

### 5.4. Genetic Engineering

One cutting-edge method for functionalizing cells involves the use of gene editing tools to selectively alter the expression of proteins on the cell surface. By genetically modifying cell lines, unique functional antigens from the source cells can be upregulated or non-native antigens can be induced, thereby adding new functionalities. Once modified, these cells can be cultured in bulk, and their genetically modified membranes can be isolated for subsequent coating on nanosystems. Since this approach integrates functionalization within the cells themselves, it has the potential to reduce the cost of large-scale production. To ensure effective gene modification, the DNA- or mRNA-based gene modifier must be able to access the cytosol in a viable form. To achieve successful intracellular delivery, a variety of methods are available, including viral vectors (such as lentivirus, adenovirus, and adeno-associated virus), chemical methods (such as cationic lipids and polymers), and physical methods (such as electroporation, microinjection, and gene gun). The ideal gene delivery technique is selected based on its feasibility, transfection efficiency, and target specificity requirements. Some studies employing genetically engineered membranes for nanosystem coating are discussed below.

Using in vitro genetic engineering to induce the expression of highly specific affinity ligands on HepG2 cells, Lv et al. developed a membrane-coated oncolytic adenovirus system. Coating with the HepG2 membrane decreased the immunogenicity of entrapped adenovirus, and overexpression of active preS1 ligands facilitated enhanced tumor accumulation. In the same study, the researchers used in-body CRISPR technology to express a small peptide, Asn-Gly-Arg, on erythrocyte membranes. This peptide was then coated onto oncolytic adenoviruses to target aminopeptidase N (APN), a cancer-specific isoform of membrane metalloproteinase. The second system also significantly enhanced tumor accumulation in APN receptor-expressing tumors, such as PC13 and U87 (Figure 10B) [181]. In a recent study, Krishnamurthy et al. explored the expression of the proline-alanine-serine (PAS) peptide sequence on HEK293 cells using a plasmid vector (encoding PAS repeats with a C-terminal transmembrane anchoring domain). The genetically engineered membrane was then coated on polymeric cores. The resulting nanosystem, called PASylated nanoghosts, reduced protein adsorption and macrophage uptake by about 90% compared to a control with a wild-type HEK293 membrane coating. The PASylation approach also extended the in vivo circulation half-life (to 37 h, a 3-fold increase compared to the control group) [182].

## 6. Theranostics Applications

### 6.1. Tumor Diagnosis and Bio-Imaging

Membrane-coated nanosystems have emerged as promising tools for tumor diagnosis and in vivo tumor imaging due to their unique physicochemical features [2]. Regardless of the source cell, the membrane camouflage facilitates the NP core to localize in close proximity to the tumor microenvironment. Additionally, the versatility of the core endows the flexibility to incorporate different types of contrast agents, such as paramagnetic agents, superparamagnetic iron oxide, metal-based NPs, and near-infrared (NIR) fluorescent nanomaterials. The following section discusses some of the latest examples of tumor diagnosis and imaging using membrane-coated nanosystems.

CTCs are cancer cells that have detached from the primary tumor and entered the bloodstream, allowing them to potentially spread to other parts of the body and form new tumors in a process known as metastasis. Detecting CTCs is crucial as it provides valuable insights into cancer progression, which can aid in making informed decisions about treatment. However, the rarity of CTCs (1–10 CTCs/mL of whole blood) in the bloodstream makes their detection a challenging task [183]. To address this, Ding et al. developed a CTC isolation/detection system by combining a multivalent aptamer-functionalized core with a cancer-WBC hybrid membrane. The core comprised magnetic Fe_3_O_4_@SiO_2_ NPs to facilitate the biotin interaction-mediated surface attachment of tetravalent-DNA-modified Ag_2_S nanodots (NIR fluorescence properties), and the hybrid cell membrane was modified with streptavidin. The resulting “nano-bioprobe” demonstrated increased affinity towards CTCs through homotypic binding and reduced interference from white blood cells. In mixed cell samples containing 500 MCF-7 cells and 106 RAW264.7 cells, the platform achieved a capture efficiency of 97.63%, surpassing uncoated bioprobe (No-M), pure CCM-coated bioprobe (CM), and pure WBC membrane-coated bioprobe (WB). The nano-bioprobe showed the highest capture purity of 96.96% due to its specificity towards CTC. The hybrid membrane coating facilitated excellent average recovery rates for spiked cells (MCF-7 cells) in lysed blood (96.24%) and whole blood (90.25%) (Figure 11A) [183].

In a related investigation, Huang et al. developed a surface plasmon resonance (SPR)-based sensing platform using gold NPs (AuNPs) functionalized with CTC membrane fragments (M-AuNPs). The CTC membrane fragment expresses junction plakoglobin (JUP), a specific cytoplasmic component. The authors employed a sensor chip customized with anti-JUP to trap M-AuNPs, which were detected via a change in SPR angle. Furthermore, by utilizing the overexpression of folic acid (FA) receptors on the CTC surface, FA-AuNPs were bound to M-AuNPs to provide dual selectivity, ensuring the sensor’s reliability and sensitivity. This approach resulted in multi-signal amplification that allowed the platform to achieve a low detection limit of 1 cell within the linear range of 10–105 cells/mL [184].

Metabolic glycan labeling is a technique that involves the incorporation of chemically modified sugars into the glycans of living cells. This technique has gained attention in tumor diagnostic imaging as it allows for selective targeting and visualization of specific tumor-associated glycans. However, tumor heterogeneity (including tumor subtypes and inter-patient heterogeneity) and the limited sensitivity of current imaging technologies are major challenges for clinical applications. With this in mind, Liu et al. reported a metal-organic framework-azidosugar complex (ZIF-8-Ac4GalNAz; ZGM) camouflaged with a cancer cell membrane. ZGM targeted homotypic cancer cells by utilizing receptors for cell-selective glycan labeling. Cellular internalization was mediated by a cholesterol-dependent endocytosis pathway, whereas the “proton sponge” effect facilitated the accelerated release and lysosomal escape. The treated cells showed significant metabolic labeling within 12 h, eliminating the need for pre-incubation. Additionally, the membrane coating provided anti-phagocytosis by macrophages and extended blood circulation, leading to efficient MGL. In vivo experiments demonstrated that the platform enabled visualization of multiple tumor cell-selective glycans on homotypic tumors. Further studies showed that the biomimetic nanosystem could differentiate between triple-negative (MDA-MB-231 cell) and luminal A subtypes (MCF-7 cell) of breast cancer. The imaging selectivity of homotypic subtypes was 5.5 times higher than that of heterotypic subtypes (Figure 11B). This level of selectivity is clinically valuable for differential diagnosis within the cancer subtype [185].

In a similar MGL-based study, Liu et al. designed pH-responsive azidosugar liposomes. The authors incorporated the cancer cell membrane as a liposome component to reduce protein corona formation and prevent phagocytosis by macrophages. This extended the blood circulation time and allowed homotypic targeting to achieve tumor localization. Compared to the single-ligand targeting strategy, the cell membrane approach resulted in approximately 1.7-fold higher glycan labeling. In addition to tumor-selective glycan imaging of different cancer cell subtypes, the platform facilitated the detection of lung metastases. In vivo experiments in both athymic nude mice and immune-competent mice showed that the imaging efficiency was over 3.4-fold higher than that obtained from conventional azidosugar-loaded liposomes [186].

**Figure 11 pharmaceutics-15-01677-f011:**
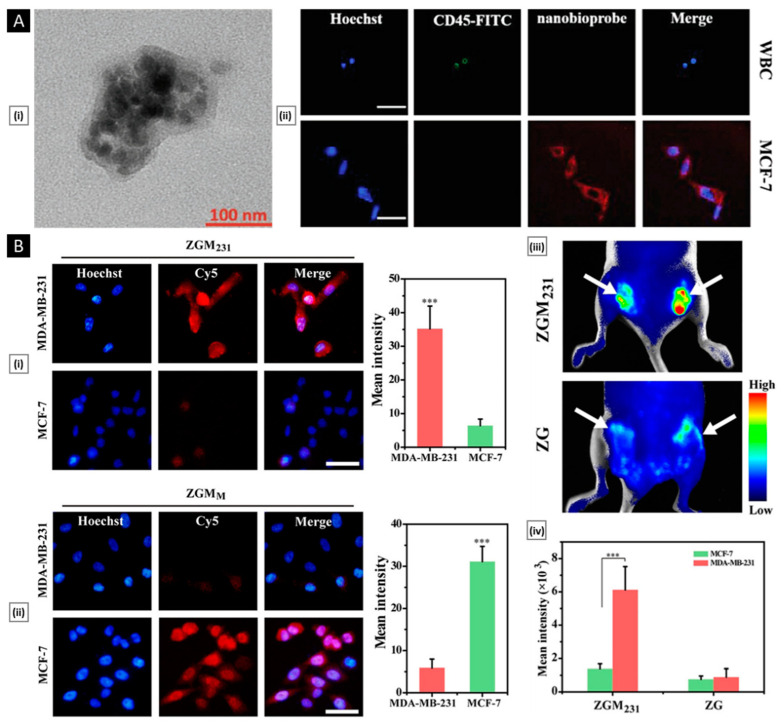
(**A**) (**i**) TEM image of CTC membrane-coated Fe_3_O_4_@SiO_2_ NPs (scale bar: 100 nm). (**ii**) Microscopic images of cells captured from mimicked clinical blood samples and identified with three-color immunocytochemistry. The merged image shows the overlapping of nuclei (Hoechst), CD45-FITC, and NIR nanobioprobe-cultured cells (scale bar: 50 µm). Reproduced with permission from [183], copyright Wiley-VCH 2020. (**B**) (**i**,**ii**) represent fluorescence micrographs and corresponding intensity analysis of different breast cancer subtypes treated with 50 µM of ZGM231 (MDA-MB-231 cell membrane-coated) and ZGMM (MCF-7 cell membrane-coated), respectively (scale bar: 50 μm). (**iii**) Whole-body fluorescence images of tumor-bearing mice injected with ZGM231 (60 mg/kg) and ZG (coating-free, 60 mg/kg) for 4 days. The white arrows marked the position of tumors (right flank: tumor, left flank: MCF-7 tumor). (**iv**) Quantification of the fluorescence signal after whole-body imaging (*** *p* < 0.001). Reproduced with permission from [185], copyright Wiley-VCH 2021.

Considerable advancements have been achieved thus far in using fluorescence imaging for in vivo diagnosis of gliomas during the preoperative stage. Nevertheless, a few obstacles still exist, such as the precise identification of cancer metastases as well as enhancing spatial and temporal resolution. The inability of nanosystems to cross the blood–brain barrier (BBB) restricts the imaging contrast’s effectiveness in detecting gliomas. To evade this drawback, Men et al. developed glial cell (C6 cell line) membrane-coated conjugated polymer dots (Pdots-C6). The nanosystem core comprised a NIR-II-absorbing polymer (a donor–acceptor complex of triphenylamine-functionalized phenothiazine with benzothiazole) for high-contrast NIR-II fluorescence imaging. The fluorescence spectrum of Pdots-C6 exhibited a strong emission peak at 1055 nm upon excitation at 808 nm. At the same excitation, the NIR-II fluorescence quantum yield was determined to be 0.6% (using IR-26 as a reference). Using a hippocampal orthotopic glioma model, the authors injected Pdots-C6 through the tail vein and monitored the fluorescence signal at the tumor site. By displaying a “do not eat me” signal to macrophages, the platform efficiently avoids systemic clearance and exhibits high brain accumulation. The C6-membrane facilitates BBB crossing ability with enhanced fluorescence signals detectable up to 48 h post-injection (Figure 12A) [187].

Following a related thought, Wang et al. employed lanthanide-doped NPs coated with brain tumor cell membranes to facilitate brain tumor imaging and surgical navigation. This innovative system demonstrated exceptional stability, high temporal and spatial resolution, and lower background signals. Cumulatively, these features enabled clear visualization of the brain tumor boundary. As a proof-of-concept, the glioma tissue (measuring less than 3 mm in size and with a depth greater than 3 mm) was clearly visualized and completely removed with the guidance of NIR-IIb fluorescence [188]. Ultrasmall iron oxide NPs (USIO NPs) have gained immense popularity as magnetic resonance imaging (MRI) contrast agents due to their superior biocompatibility. If their size is <5 nm, they exhibit reduced magnetization via the spin-canting effect, which allows them to efficiently shorten the T1 relaxation time of water protons, enhancing T1-weighted MRI. On the contrary, a size > 5 nm generally shows T2 MR signals [189]. To enhance their tumor accumulation, Jia et al. crosslinked multiple 3.2 nm USIO NPs by using a pH-responsive benzoic imine-based inter-particle crosslinking strategy, which was subsequently encapsulated within the cancer cell membrane. When crosslinked, the aggregated nanosystem exhibits strong T2 MRI signals. However, upon exposure to the acidic pH of the tumor microenvironment, USIO NPs disintegrate into individual particles and stimulate a dynamic T2/T1 switchable signal. The tumor MR imaging could be further synergized via the ultrasound-induced sonoporation effect (Figure 12B) [190].

Another interesting class of molecular probes for tumor bioimaging are upconversion NPs (UCNPs). They are optical nanomaterials (usually doped with lanthanide ions) with the unique feature of converting near-infrared excitation into visible and ultraviolet emission. In addition, UCNPs have a long luminescence lifetime and high photostability, which enable long-term imaging and repetitive scanning [191]. Fang et al. developed biomimetic nanoprobes by combining cancer cell membranes with Gd^3+^-doped UCNPs for multimodality (upconversion luminescence with MRI) and precise imaging of triple-negative breast cancer (TNBC). The nanosystem showed reduced immunogenicity and homologous targetability was assessed in MDA-MB-231 cell lines by capturing the upconversion luminescence (post-irradiation with a 980 nm laser). The in vivo fluorescence-based imaging studies in immunocompromised BALB/C nude mice highlighted the platform’s target-specific biodistribution and improved imaging for early diagnosis of highly aggressive TNBC. The platform facilitated enhancement in tumor-specific T1-weighted MR images. Additionally, it possessed excellent biosafety and did not cause any deaths or organ damage after 30 days of administration [192].

Erythrocyte-derived membranes have emerged as a promising candidate for tumor imaging in addition to cancer cells. This is due to their ability to remain undetected by the immune system and their longer circulation time, enabling them to efficiently reach tumors located at distant sites. Li et al. combined UCNPs with erythrocyte-derived membranes. Tumor uptake was enhanced by incorporating 1,2-distearoyl-sn-glycero-3-phosphoethanolamine-N-[folate(polyethylene glycol)-2000] (DSPE-PEG-FA) molecules onto the cell membranes. In addition to MRI and UCL imaging, the developed platform facilitated positron emission tomography (PET), which can overcome the deficiencies of previous approaches in detecting deep tissues. Using a pre-targeting strategy and “in vivo” click chemistry, the authors performed PET imaging in 4T1 breast cancer-bearing mice (Figure 12C). Results of blood biochemistry profiling, hemocompatibility assays, and histologic analysis suggested good in vivo biocompatibility of the reported nanosystem [193]. Ferrel et al. designed a biomimetic liposome (BML) by re-engineering synthetic liposomes with the membranes of erythrocytes for diagnostic applications. The liposomal component consisted of biodegradable synthetic phospholipids such as 1,2-distearoyl-sn-glycero-3-phosphoglycerol, 1,2-distearoyl-sn-glycero-3-phosphoethanolamine, and a gadolinium-conjugated lipid (which served as an MRI contrast agent). The final BML platform, formed by fusing liposome vesicles with erythrocyte membranes via membrane extrusion, had a hydrodynamic diameter of 180 ± 20 nm and a negative surface charge of 29 ± 2 mV. The longitudinal relaxivity (r1) of BML is 3.71 mM^−1^s^−1^, which is similar to the r1 of the commercial contrast agent, Magnevist [194].

**Figure 12 pharmaceutics-15-01677-f012:**
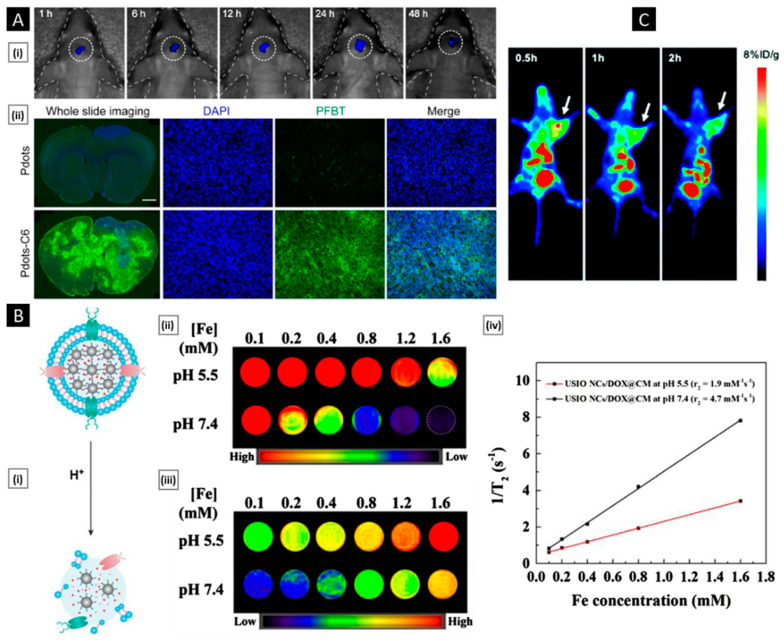
(**A**) (**i**) In vivo NIR-II fluorescence imaging of gliomas with Pdots-C6 administration at different time points post-injection. (**ii**) Brain tissue section 24 h post-injection. Green: PFBT; blue: cell nuclei stained with DAPI (scale bar: 1 mm). Reproduced with permission from [187], copyright Elsevier 2022. (**B**) (**i**) Schematic illustration of the dissociation of cancer cell membrane-coated USIO NPs in an acidic environment. (**ii**,**iii**) represent T2-weighted and T1-weighted MR images of cancer cell membrane-coated USIO NPs, respectively. (**iv**) Plots of 1/T2 as a function of Fe concentration. Reproduced with permission from [190], copyright Elsevier 2021. (**C**) Time-dependent PET imaging of 4T1 breast cancer-bearing mice injected with erythrocyte-membrane-coated UCNPs. Reproduced with permission from [193], copyright RSC Publishing 2020.

### 6.2. Anti-Tumor Therapy

Knowledge regarding the biology and functioning of cancer has been evolving rapidly, leading to the identification of several drugs that can be effectively utilized in antitumor therapy. These advancements have opened up new possibilities for the development of targeted and personalized treatments that can improve patient outcomes and quality of life. Researchers and clinicians alike have been examining the potential of innovative approaches such as membrane-coated nanosystems for combating cancer in recent years. The following section highlights some recent studies focused on antitumor therapy by delivering anticancer biomolecules, photothermal/photodynamic agents, and cancer immunomodulators.

Multiple myeloma (MM) is a hematological malignancy that occurs due to the uncontrolled proliferation of malignant plasma cells in the bone marrow (BM), which causes bone destruction, hematopoiesis inhibition, and overcrowding of normal plasma cells. It is the second most prevalent hematological malignancy, accounting for 15% of all hematological neoplasms worldwide and significantly impacting patient quality of life. Bortezomib (BTZ), a proteasome inhibitor, is the first-line clinical agent for MM and can regulate the BM microenvironment, target MM cells, and reduce bone damage. Although chemotherapy with BTZ has benefits, it is not curative since delivering an adequate amount of the drug to the primary MM site is difficult, which fails to eradicate MM cells within the BM. Exploiting the fact that MM cells migrate and return to the BM for survival and proliferation (a phenomenon that is termed “bone marrow homing”), Qu et al. prepared BTZ-loaded polymeric NPs and subsequently coated them with MM cell membrane via physical extrusion (MPCEC@BTZ NPs). In a systemic orthotopic transplantation MM model, the nanosystem demonstrated remarkable antitumor efficacy. This can be attributed to BM localization facilitated by the presence of BM-specific protein markers such as CD44, CD147, CXCR4, and CD138. The platform delayed tumor progression in the BM, improved overall survival, and reduced systemic side effects with no histological toxicity in any major organs (Figure 13A) [195].

Photothermal therapy (PTT) is a non-invasive cancer treatment approach that utilizes light-absorbing agents, known as photothermal agents, to induce localized hyperthermia. The photothermal agent is administered to the patient either intravenously or topically, depending on the location of the cancerous cells. Subsequently, a laser emitting light in the NIR range illuminates the tumor site, which the photothermal agent absorbs. The absorbed light energy is then converted into heat energy, leading to hyperthermia that induces cellular damage and ultimately causes cancer cell death [196]. Although magnetic iron oxide NPs are excellent photothermal agents, early macrophage-mediated clearance and poor tumor localization limit their utility. Yuan et al. reported the 4T1 tumor exosome-coated nanosystem (Fe_3_O_4_@Exo NPs) as a NIR-sensitive drug delivery system for synergistic chemo-photothermal co-therapy. DOX was used as a model drug that was loaded onto Fe_3_O_4_ NPs via the incubation method. Exosome coating was achieved via sonication, followed by repeated co-extrusion of NPs with exosome fragments. Upon irradiation for 270 s with an 808 nm laser (2 W/cm^2^), the system reaches a temperature of around 46 °C, which collapses the external exosome coating, causing the rapid release of entrapped DOX. The cumulative release percentages of DOX at pH 7.4, 6.0, and 5.0 media at 6 h were found to be 35.41, 44.30, and 46.44%, respectively. Such pH sensitivity is advantageous in preventing premature release of DOX in bodily fluids and promoting release in the slightly acidic conditions of tumors. As a naturally occurring endogenous transport carrier, exosomes shield Fe_3_O_4_ NPs from being engulfed by the mononuclear phagocyte system. Additionally, PTT-induced hyperthermia potentiated DOX, as evident by the robust suppression of tumor growth (vs. only PTT or only DOX group) (Figure 13B) [197].

In a similar exosome-based study, Zhang et al. utilized exosomes sourced from murine brain endothelial cells (bEnd.3) to promote BBB penetration of DOX-loaded polymeric NPs. The nanosystem (termed ENPDOX) expresses proteins relevant to exosomes, including CD9, CD63, CD81, and HSP70. In vitro BBB assays showed that ENPDOX effectively penetrated mouse endothelial cells and induced apoptosis in glioma cells. Notably, ENPDOX treatment resulted in increased surface calreticulin exposure, HMGB1 secretion, and ATP release, which are hallmarks of chemotherapy-induced immunogenic cell death. In an in vivo mouse model, ENPDOX administration via intravenous injection led to DOX accumulation in the glioblastoma xenograft. The use of ENPDOX significantly increased the maturation of DCs in cervical lymph nodes, activated cytotoxic T lymphocytes, and promoted the production of proinflammatory cytokines [198].

Photodynamic therapy (PDT) is an intriguing treatment approach for tumors that involves using a light-excitable photosensitizing agent to generate reactive oxygen species (ROS) such as singlet oxygen (^1^O_2_), superoxide radicals (•O^−^_2_), and hydroxyl radicals (•OH). When directed at tumors, these ROS can induce irreversible damage, stimulate an immune response, and deform tumor microvasculature, which cumulatively causes cancer cell apoptosis or necrosis [199]. Several studies have reported PDT using membrane-coated nanosystems. Zhang et al. developed neutrophil membrane-coated poly(lactic-co-glycolic acid) (PLGA) NPs for PDT of hepatocellular carcinoma. Hypocrellin B, a pigment obtained from the fungus Hypocrella bambusae, was employed as the photosensitizer. Activated neutrophils were obtained from an animal model with LPS-induced inflammation. The resulting nanosystem, designated as NM-HB NPs, exhibited an average diameter of 140 nm and a zeta potential of −24.8 mV in water. The NM-HB NPs were selectively targeted to invasive cancer cells and were found to effectively down-regulate cytokines in the inflammatory tumor microenvironment (vs. neutrophil membrane-free NPs). The deep tumor penetration ability of the platform was confirmed through fluorescence imaging of tumor-bearing mice 12 h post-intravenous administration. It was also observed that induction of photodynamic therapy efficiently decreased serum TNF-α and IL-6. However, no such results were observed when no laser treatment was given. This suggests that treatment with NM-HB NPs along with PDT could effectively inhibit hepatocellular carcinoma growth. Histological analysis of Kunming mice confirmed that neutrophil-camouflaged NPs did not induce any damage or inflammation, indicating their safety [200].

In a recent study, Pan et al. developed a cancer cell membrane-coated platform with a photosensitizer-based nanoscale MOF core for combined PDT with chemotherapy for breast cancer. The core was comprised of polyvinylpyrrolidone-dispersed Fe-tetrakis (4-carboxyphenyl) porphyrin loaded with the hypoxia-activable prodrug tirapazamine (TPZ). The functional concept of the nanosystem can be explained through the following sequence of events: The cancer cell membrane coating helps in evading immune clearance and allows for tumor retention. Upon reaching the tumor lesion, the nanosystem undergoes endocytosis and enters the cells. It then decomposes within the lysosomes, releasing Fe^3+^ due to its acid-responsive property. The Fe^3+^ catalyzes the endogenous H_2_O_2_ to produce •OH and lower glutathione levels, thereby modulating the inherent TME and triggering ferroptosis. The increased oxygen content in the TME enhances the PDT effect, leading to cell apoptosis. Finally, the PDT process consumes a significant amount of oxygen, inducing serious hypoxic conditions that activate the prodrug TPZ for chemotherapy (Figure 13C). The designed nanosystem (termed PFTT@CM) had a hydrodynamic diameter of 201 nm and a zeta potential of −44.22 mV. The drug loading efficiency of TPZ was found to be 27.1 ± 7.4%. Based on the cell viability data, the proportion of the therapeutic effects of ferroptosis, PDT, and TPZ-based chemotherapy against MDA-MB-231 cells by PFTT@CM (100 μg/mL) was roughly estimated to be 19.7%, 43.8%, and 22.9%, respectively. The hypoxic microenvironment was visualized using a cellular ROS and hypoxia stress detection kit, which confirmed the robust activation of TPZ. Following this, the TPZ-mediated nucleic acid damage, responsible for causing functional incapacitation of cancer cells, was evaluated using the TUNEL assay. As expected, the cumulative effect of PDT, TPZ, and ferroptosis caused the highest anticancer effect (vs. any individual therapy). In vivo biodistribution was assessed in tumor-bearing nude mice via real-time fluorescence imaging. PFTT@CM showed preferential accumulation in tumors with significant fluorescence intensity, even 96 h post-injection. The synergistic antitumor activity led to complete suppression of the tumor for 18 days after a single treatment of PFTT@CM [201].

**Figure 13 pharmaceutics-15-01677-f013:**
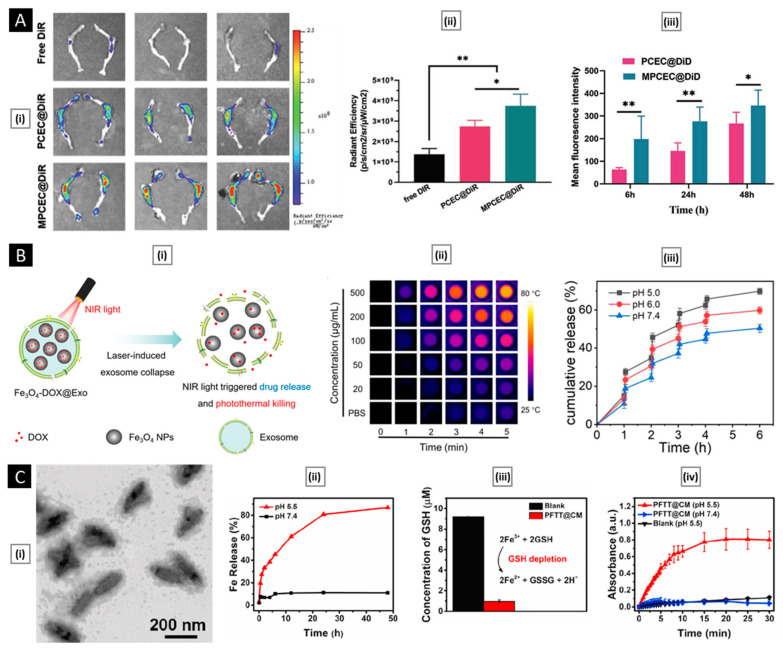
(**A**) (**i**) Ex vivo fluorescence imaging of the femur tissues at 48 h post-injection of MPCEC@Dir in the 5T multiple myeloma (5TMM) murine model. (**ii**) Quantitative analysis of the radiation efficiency of the femur tissues at 48 h post-injection. (**iii**) Quantitative analysis of fluorescence intensity in the bone marrow through flow cytometry analysis after injection of the DiD-labeled NPs (* *p* < 0.05, ** *p* < 0.01). Reproduced with permission from [195], copyright Wiley-VCH 2021. (**B**) (**i**) Schematic illustration showing NIR-triggered drug delivery and PTT therapy using Fe_3_O_4_-DOX@Exo NPs. (**ii**) Infrared thermographic images of Fe_3_O_4_-DOX@Exo dispersions after irradiation for 5 min. (**iii**) Cumulative release profiles of DOX in PBS at pH 5.0, 6.0, and 7.4 in the presence of NIR irradiation for 3 min. Reproduced with permission from [197], copyright American Chemical Society 2022. (**C**) (**i**) TEM images of PFTT@CM. (**ii**) Cumulative Fe^3+^ release from PFTT@CM at pH 5.5 and pH 7.4, showing its acid-dependent dissociation. (**iii**) GSH depletion ability of PFTT@CM mediated by Fe^3+^ (**iv**) Continuous H_2_O_2_ catalyzation by PFTT@CM as monitored by the 3,3′,5,5′-Tetramethylbenzidine assay (in 30 min). Reproduced with permission from [201], copyright Elsevier 2022.

Immunotherapy harnesses the immune system to fight cancer and is a rapidly developing field in biomedical research. It uses tumor antigens and other tumor-associated macromolecules to train the body’s immune system to recognize and target cancer cells. Compared to traditional cancer treatments, immunotherapy is safer and more effective. Mainstream immunotherapy-based treatments include immune checkpoint blockers and CAR-T therapy, while cancer vaccines with molecular adjuvants are also being explored [202]. However, non-specific inflammation, autoimmune-like disorders, and severe toxicity-related disorders are critical drawbacks of immunotherapy. Membrane-coated nanosystems can address these issues by precisely targeting the tumor immune microenvironment with optimal source cells and immunomodulators.

It has been observed that under neuroinflammatory conditions, immune cells cross the BBB via the endothelial cell body (through pores) and brain barrier junctions. Rapamycin (RAPA) is a powerful mTOR inhibitor and a strong autophagy inducer with potential applications in countering aggressive gliomas. To facilitate its BBB transfer, Ma et al. designed an activated mature DC membrane-coated nanosystem (aDCM@PLGA/RAPA). The femoral and tibial bones of 6- to 8-week-old male C57BL/6 mice were used to isolate DCs, followed by their activation (using GM-CSF and IL-4) and modification with tumor cell lysate (TCL) protein (to enhance glioma cell uptake). The use of activated DC membrane boosts immunotherapy as it allows enhanced antigen presentation to activate native CD8^+^ T cells. The activity of RAPA, when combined with the activation of T cells, results in the infiltration of NK cells in the peripheral immune organ, which imparts significant anti-glioma efficacy. The platform had an average diameter of 127.9 ± 1.5 nm, a zeta potential of −23.2 ± 1.0 mV, and a RAPA encapsulation efficiency of 57.55%. Using an in vitro transwell assay, BBB penetration was evaluated, which highlighted the efficiency of the TCL protein-bearing DC cell membrane. Lastly, in vivo BBB-crossing evaluation and antitumor studies were carried out via an orthotopic glioma mouse model (using luciferase-labeled C6 cells). The fluorescence intensity of the treatment group (DiR-loaded nanosystem) was roughly 7-fold higher than that of free DiR dye. Treatment with aDCM@PLGA/RAPA not only significantly inhibited glioma growth but also had the potential to induce glial differentiation in the orthotopic glioma. Additionally, aDCM@PLGA (without RAPA) induced robust CD8^+^ activation, significantly enhanced the secretion of immune stimulatory cytokines (IL-2, IFN-γ, TNF-α, and IL-6) and suppressed orthotopic glioma growth in a prophylactic setup, suggesting its potential use as a cancer vaccine to induce tumor immunity (Figure 14A) [203].

In glioblastoma, the immune-suppressing tumor microenvironment causes microglia and macrophages to polarize into a tumor-promoting M2 type, which hinders antigen presentation and T cell proliferation, promoting tumor cell growth and invasion. Current microglia/macrophage-based immunotherapy strategies mainly aim at their elimination, which may compromise their beneficial functions such as phagocytosis and antigen presentation [204]. Hence, regulating microglia/macrophage phenotypes is a clinically advantageous strategy. To achieve this, Gao et al. developed a novel nanosystem termed Vir-Gel, comprising virus-mimicking membrane-coated nucleic acid nanogels loaded with miR155. The system was constructed by first preparing a miR155-loaded DNA-grafted polymer core (nanogel), which was then coated with an erythrocyte membrane to prolong its circulation time. Two influenza virus-derived peptides, M2pep and HA2, were further functionalized onto the membrane-coated nanogel. M2pep specifically targeted M2 microglia, enabling the targeted delivery of the miR155-loaded Vir-Gel. The HA2 peptide promoted the fusion of the erythrocyte membrane with the microglia membrane, leading to enhanced internalization. miR155 was released in response to ribonuclease H and remodeled the M2-like microglia into M1-like microglia, leading to the reversal of immunosuppressive TME to immunogenic TME. In vitro studies showed that treatment with Vir-Gel significantly increased the production of specific markers of M1-like microglia, such as inducible nitric oxide synthase (iNOS), IL-6, and TNF-α. Parallelly, the expression of CD206, a marker of M2-like microglia, was decreased. Live imaging after systemic injection demonstrated a high accumulation of Vir-Gel at the glioma site. Furthermore, treatment with Vir-Gel prolonged the survival time of glioma-bearing mice (from 15.6 to 27.6 days vs. the saline-treated group) and led to a much higher percentage of M1-like microglia within the glioma TME (Figure 14B) [205].

Immunogenic cell death (ICD) is a crucial aspect of immunotherapy that can enhance the antigenicity and adjuvanticity of tumor sites, thereby activating the immune microenvironment and rendering tumors more receptive to immune checkpoint inhibitor intervention. Li et al. reported a self-amplified nanosystem called mEHGZ. This nanosystem comprises epirubicin (EPI), glucose oxidase (Gox), and hemin encapsulated in a MOF-based NP and is further coated with the calreticulin-overexpressed tumor cell membrane. This biomimetic delivery system induces ICD by using EPI as an ICD inducer, Gox and hemin to generate ROS (which strengthens the ICD effect), and calreticulin-rich membrane as an “eat me” signal to promote the presentation of antigens by DCs (to invoke the tumor-immunity cycle). The delivery system exhibits an amplified ICD effect through Gox oxidation, hydroxyl radical generation, and glutathione depletion. The potent ICD effect promotes DC maturation and infiltration of cytotoxic T lymphocytes, thereby reversing an immunosuppressive tumor microenvironment to an immunoresponsive one. Treatment with this nanosystem in combination with an anti-PD-L1 antibody results in significant inhibition of tumor growth and lung metastasis. This indicates that a strong ICD effect can significantly enhance the therapeutic efficacy of anti-PD-L1 antibodies [151].

**Figure 14 pharmaceutics-15-01677-f014:**
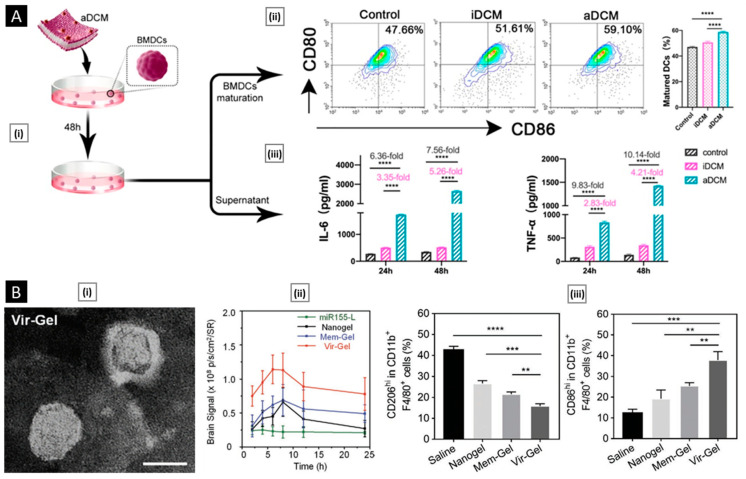
(**A**) (**i**) Illustration of in vitro DCs activation. (**ii**) Quantitative analysis of the expression of CD80/86 to assess DCs maturation. (**iii**) Detection of IL-6 and TNF-α in DCs suspension by ELISA kit (**** *p*< 0.0001). Reproduced with permission from [203], copyright American Chemical Society 2023. (**B**) (**i**) Representative TEM image of Vir-Gel (scale bar: 100 nm). (**ii**) Quantification analysis of the fluorescent signals in the brain area. (**iii**) Relative quantification of M2 microglia and macrophages and M1 microglia and macrophages in the tumor tissues. (** *p* < 0.01, *** *p* < 0.001, **** *p*< 0.0001). Reproduced with permission from [205], copyright Wiley-VCH 2021.

An emerging concept in the field of antitumor therapy is the combination of PTT with immunotherapy. PTT can increase the antigenicity and adjuvanticity of tumor sites by promoting the release of tumor-related antigens and damage-associated molecular patterns and by increasing the expression of major histocompatibility complex class I (MHC-I) molecules on dying tumor cells, which sensitizes CD8^+^ T cells. PTT-induced hyperthermia can further stimulate cytokine production such as IFN-γ and TNF-α, which in turn promote the activation and tumor infiltration of T cells [206]. Several studies have relied on membrane-coated nanosystems for the co-delivery of a photothermal agent with an immunomodulator.

In a recent study, Yang et al. recently developed a nano-vaccine composed of cancer cell membrane vesicles encapsulating manganese dioxide NPs and DiR, a photothermal agent. The cancer cell membrane served as a source of tumor antigens and helped in homotypic targeting. By releasing Mn^2+^, manganese dioxide NPs can stimulate STING (stimulator of interferon genes) responses. Hydrogen peroxide and hydronium ions rapidly degrade the NPs, restoring pH levels back to normal and generating oxygen to alleviate tumor hypoxia. The remaining DiR-loaded membrane vesicles were used for photothermal therapy, enhancing tumor killing and antigen release. The NPs were most effective when administered intravenously and combined with laser irradiation in a primary tumor model. This resulted in significant infiltration of CD4^+^ and CD8^+^ T cells at the tumor site and a low number of CD4^+^CD25^+^Foxp3^+^ regulatory T cells. In a surgical resection recurrence model, the nanosystem was applied to large tumors before the complete removal of the primary tumor. After 12 days, when a secondary tumor was induced on the contralateral flank, three out of five mice treated with the nanosystem and PTT did not show tumor growth. Similar results were obtained in a bilateral tumor model where only the primary tumor was subjected to laser treatment (Figure 15A) [207].

Following a similar strategy, Xiong et al. utilized a hybrid cell membrane to conceal indocyanine green (ICG)-loaded magnetic NPs (Fe_3_O_4_-ICG@IRM) for combined PTT-immunotherapy of ovarian cancer. They fused a murine-derived ID8 ovarian cancer cell membrane with an RBC membrane, which allowed for long circulation due to the RBC components and induced an antitumor immune response upon delivery to the spleen/lymph nodes through the tumor component. The hybrid membrane NPs exhibited 1.7-fold and 2.1-fold higher tumor accumulation, respectively, compared to NPs coated with only cancer or RBC membrane (based on fluorescence imaging at 8 h post-IV injection). When the NPs were IV injected into ID8 subcutaneous tumor-bearing mice and their tumors irradiated (808 nm laser, 1 W/cm^2^, 10 min), the tumor temperature increased to 54.3 °C. The researchers also tested Fe_3_O_4_-ICG@IRM via intratumoral injections in two types of bilateral flank tumor models (left ID8/right ID8, left ID8/right B16-F10). At 12 h post-injection, the Fe_3_O_4_-ICG@IRM were mainly found in draining lymph nodes, where an immune response could be activated by the tumor cell antigens in the hybrid membrane coating. The administration of Fe_3_O_4_-ICG@IRM into the left flank ID8 tumor with or without NIR irradiation had no inhibitory effect on the B16-F10 tumor in the right flank. However, the ID8 tumor antigens induced specific immune responses against non-irradiated ID8 tumors in the right flank after irradiation of the left ID8 tumor (Figure 15B). The researchers hypothesized that the higher accumulation of Fe_3_O_4_-ICG@IRM in the spleen activated the immune response to achieve tumor-specific immunotherapy. Additionally, the PTT-induced release of whole-cell tumor antigens into the tumor microenvironment also contributed to the antitumor immunotherapy and inhibition of primary and metastatic tumors [208].

Bacterial membrane-based immunotherapy for cancer is a promising but under-explored area that has gained attention in recent years. The main drawback of photothermal agents is their inability to trigger anticancer immunity when used systemically. To address this issue, Qin et al. developed biomimetic copper sulfide NPs (CuS-BMVs) using bacterial outer membrane vesicles (OMVs) from Escherichia coli Nissle 1917 for systemic PTT-immunotherapy against both primary and distant tumors. The OMVs in CuS-BMVs contain bacterial-derived antigens and various pathogen-associated molecular patterns (such as lipopolysaccharides, outer membrane proteins, and lipoproteins), making them effective vaccine adjuvants that activate both humoral and cellular immune responses. CuS-BMVs showed excellent photostability, tumor-targeting ability, and high photothermal conversion efficacy. Upon NIR-II irradiation at the tumor site, the nanosystem generates localized hyperthermia and induces strong ICD, DC maturation, and CD8^+^ T cell activation. CuS-BMVs also polarize M2-like tumor-associated macrophages into M1-like phenotypes, which remodels the immune-suppressive tumor microenvironment and amplifies PTT-mediated antitumor immunity (Figure 15C) [209].

## 7. Conclusions and Perspective

In summary, “bioinspired” membrane-coated nanosystems have emerged as a highly promising approach for cancer theranostics, combining the advantages of synthetic nanoparticles and natural cell membranes. The coating of nanoparticulate cores with cell membranes derived from various cell types, such as red blood cells, cancer cells, and immune cells, provides a versatile platform for targeted drug delivery and imaging applications. Techniques such as physical extrusion, sonication, and microfluidic coating have been successfully employed to coat cell membranes onto nanoparticle cores, resulting in hybrid nanosystems with improved pharmacokinetic performance and enhanced biocompatibility. The membrane coating confers immune evasion capabilities, enabling specific targeting of cancer cells while minimizing off-target effects. The characterization of membrane-coated nanosystems is crucial for assessing their stability, surface charge, and integrity. Techniques such as dynamic light scattering, zeta sizer, TEM, SEM, and CLSM provide valuable insights into their physical properties and cellular interactions. Additionally, the analysis of membrane proteins through amino acid analysis, UV–vis spectroscopy, mass spectrometry, and SDS-PAGE allows for the evaluation of their composition and activity. Furthermore, the functionalization of cell membrane-coated nanosystems offers additional possibilities beyond the properties conferred by the natural membrane source. Techniques such as lipid insertion, membrane hybridization, metabolic engineering, and genetic modification enable the incorporation of diverse functionalities. Genetic modification allows selective alteration of cell surface protein expression using gene editing tools, while metabolic engineering manipulates natural biosynthetic pathways to regulate cellular properties. Membrane-coated nanosystems exhibit great potential in tumor diagnosis and bio-imaging, leveraging the membrane camouflage for localization in the tumor microenvironment and the incorporation of various contrast agents. Moreover, they hold promise in antitumor therapy, capitalizing on advancements in cancer biology to identify drugs suitable for targeted and personalized treatments.

With this in mind, it is imperative to address several crucial factors before the clinical implementation of these innovative platforms. Although the idea of extracting membranes from diverse cell sources, coating nanoparticles with them, and reintroducing them into patients shows potential, it is vital to recognize the limitations and challenges inherent in this technology. Firstly, the process of membrane extraction and nanoparticle coating is a complex and labor-intensive procedure that involves multiple steps. It necessitates the acquisition of specific cell types from tumor patients. The extraction and purification of membranes from these cells require meticulous techniques to maintain their structural integrity and functionality. Subsequently, the membranes need to be properly coated onto nanoparticles, ensuring stability and uniformity. During the entire process, it is imperative to ensure the integrity of the cell membrane, and it is crucial to make sure the crucial membrane proteins remain viable during coating and occupy a proper orientation that allows them to efficiently perform their desired biological function.

All the abovementioned steps involve specialized equipment, skilled personnel, and substantial time investment. Moreover, the clinical application of membrane-coated nanosystems may face challenges related to scalability and cost-effectiveness. Manufacturing these nanosystems on a large scale while maintaining consistent quality and reproducibility could be demanding. The production costs, including the acquisition of cells, purification processes, and quality control measures, may contribute to the overall expense of this technology. These factors, when coupled with the lack of regulatory guidelines for using such bioinspired nanosystems, could potentially limit the widespread adoption of membrane-coated nanosystems in clinical settings. It is important to note that the use of membrane-coated nanosystems is still in the early stages of development, and further research is needed to overcome these challenges. Future advancements may explore alternative approaches to streamline the manufacturing process, reduce costs, and increase efficiency. For example, advancements in cell culture techniques, membrane isolation methods, and nanoparticle coating strategies could enhance the feasibility and practicality of this technology. Recently, several commercial cell membrane isolation kits (such as Mem-PER™ Plus from Thermo Fisher Scientific (Waltham, MA, USA), ProteoExtract^®^ from MilliporeSigma (Burlington, VT, USA), and Minute™ by Invent Biotechnologies (Plymouth, MN, USA)) are available that use proprietary filter systems to extract cell membranes (with viable surface proteins) with a high yield in a few minutes. As this field evolves, there is potential for the development of combined membrane isolation/nanoparticle coating kits that can be used in patients following proper clinical approval.

Despite the current limitations, the clinical transformation potential of bioinspired membrane-coated nanosystems should not be overlooked. This technology offers unique advantages, such as biocompatibility, prolonged circulation, and immune evasion, which are highly desirable in targeted cancer therapy. With ongoing research and innovation, it is conceivable that these nanosystems could find applications in personalized medicine, where specific cell sources from individual patients can be utilized for tailored treatments. Collaboration between researchers, clinicians, and industry partners will be crucial in addressing the challenges and facilitating the translation of this technology from the laboratory to clinical practice.

## Figures and Tables

**Figure 1 pharmaceutics-15-01677-f001:**
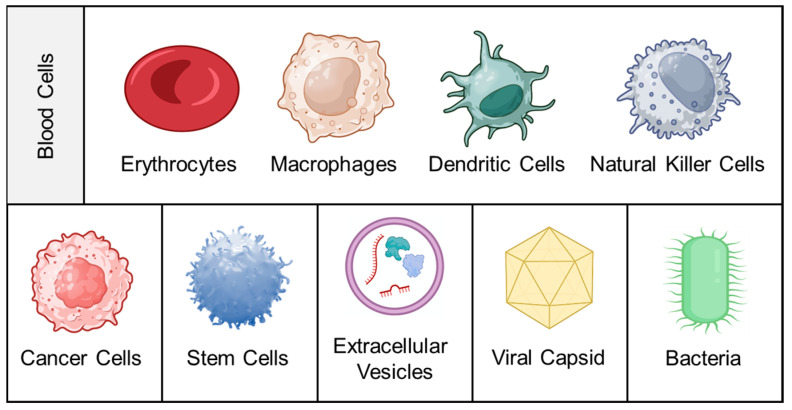
Cell membrane sources.

**Figure 2 pharmaceutics-15-01677-f002:**
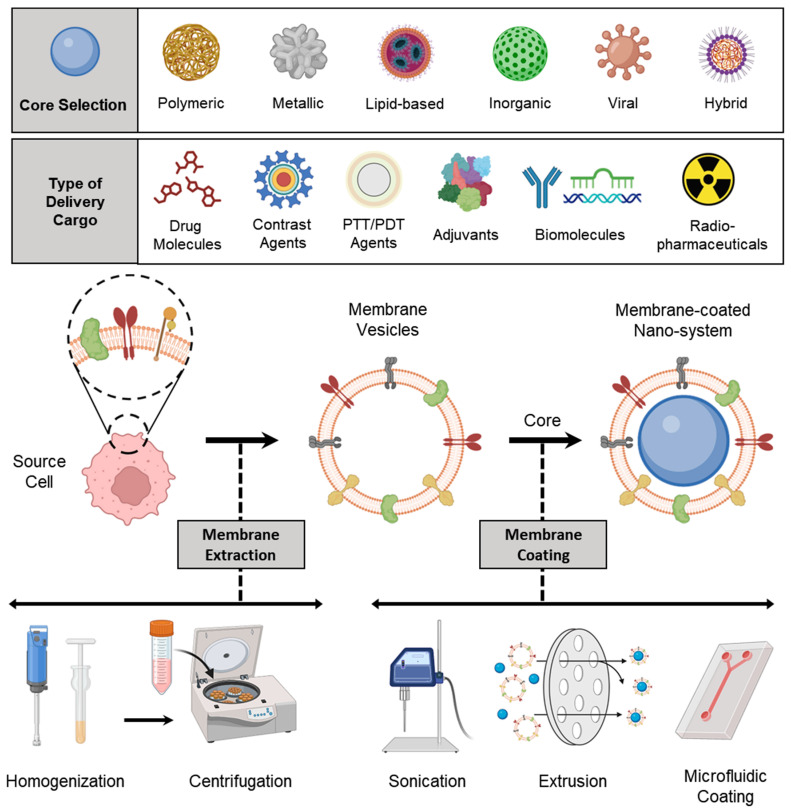
Overview of the process involved in the preparation of membrane-coated nanosystems.

**Figure 3 pharmaceutics-15-01677-f003:**
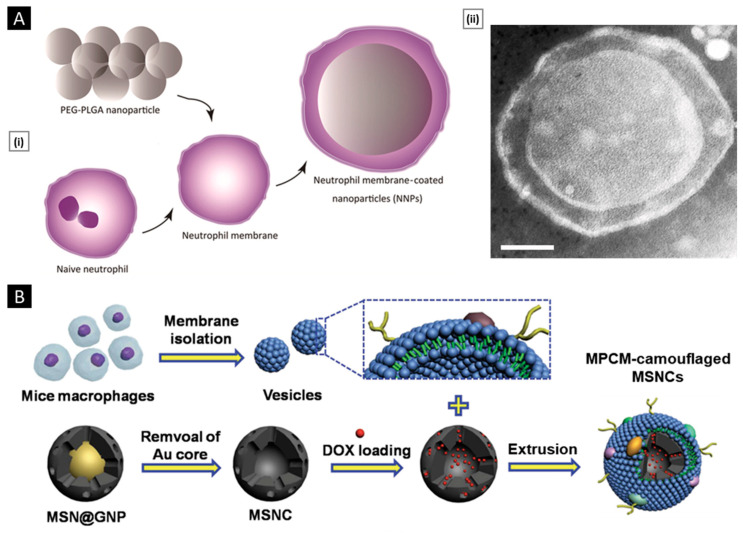
(**A**) (**i**) NNPs developed via physical extrusion for targeted drug delivery to pancreatic carcinoma. (**ii**) TEM image of NNPs. Scale bar: 30 nm. Reproduced from reference [127]. (**B**) DOX-loaded, macrophage cell membrane (MPCM)-camouflaged mesoporous silica nanocapsules (MSNCs) developed via physical extrusion method. Reproduced with permission from reference [129].

**Figure 4 pharmaceutics-15-01677-f004:**
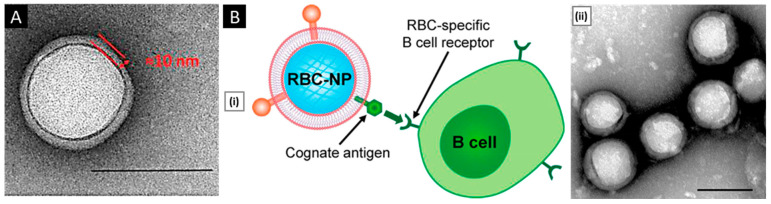
(**A**) Morphology of NM-NPs. Reproduced with permission from [132], copyright American Chemical Society 2017. (**B**) (**i**) Schematic of membrane-coated NPs for the targeting of antigen-specific B cells formed using the sonication technique. (**ii**) TEM of RBC-NPs negatively stained with uranyl acetate (scale bar: 100 nm). Reproduced with permission from [133], copyright American Chemical Society 2018.

**Figure 5 pharmaceutics-15-01677-f005:**
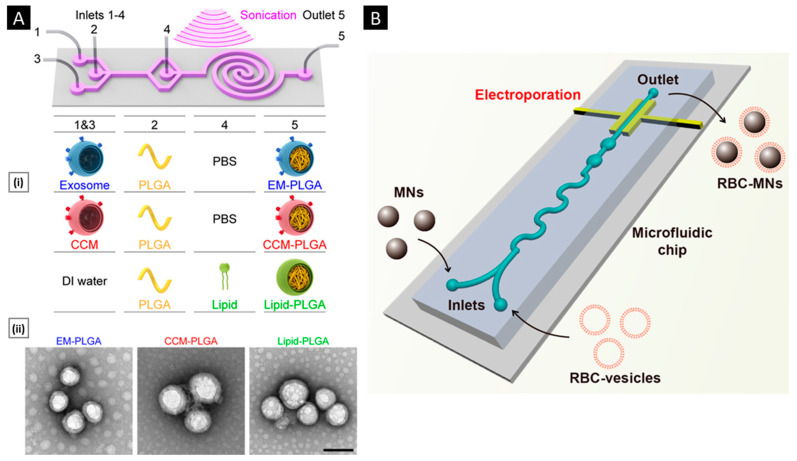
(**A**) (**i**) Schematic of the microfluidic sonication method to assemble biomimetic core–shell NPs for immune evasion-mediated tumor targeting. (**ii**) TEM characterization of EM-PLGA NPs, CCM-PLGA NPs, and lipid-PLGA NPs (scare bar: 100 nm). Reproduced with permission from [134], copyright American Chemical Society 2019. (**B**) Synthesis of RBC-MNs using microfluidic electroporation. Reproduced with permission from [135], copyright American Chemical Society 2017.

**Figure 7 pharmaceutics-15-01677-f007:**
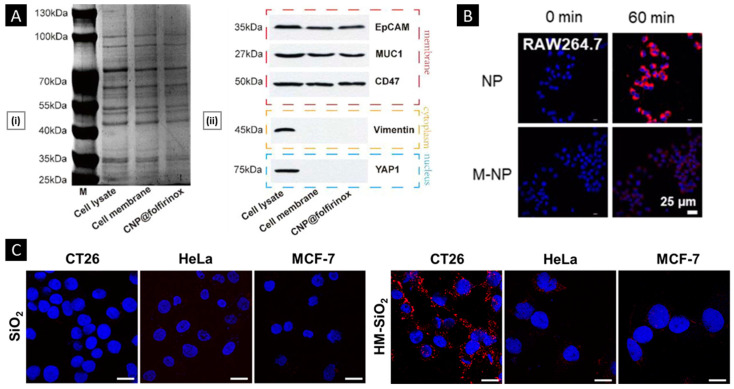
(**A**) (**i**) SDS-PAGE protein analysis of cell lysate, isolated cell membrane fragments, and membrane-coated nanosystem (CNP@folfirinox) stained with Coomassie Blue. (**ii**) Western blotting analysis of membrane-specific protein markers (absence of blots for cytoplasm and nucleus for nanosystem indicate the efficiency of membrane isolation process). Reproduced with permission from [153], copyright Elsevier 2023. (**B**) In vitro cell uptake of membrane-coated NPs (M-NPs) in RAW264.7 macrophages were detected by confocal microscopy, showing effective evasion of phagocytosis (scale bar: 25 μm). Reproduced with permission from [154], copyright Elsevier 2023. (**C**) CLSM images of three cancer cell lines (CT26, HeLa, and MCF-7) incubated with SiO_2_ NPs and HM-SiO_2_ NPs for 4 h. Preferential localization in CT26 (source cell) indicates homotypic targeting. Blue, cell nuclei stained with DAPI; red, Cy5-labeled SiO_2_ cores (scale bars: 20 μm). Reproduced with permission from [155], copyright Springer Nature 2022.

**Figure 8 pharmaceutics-15-01677-f008:**
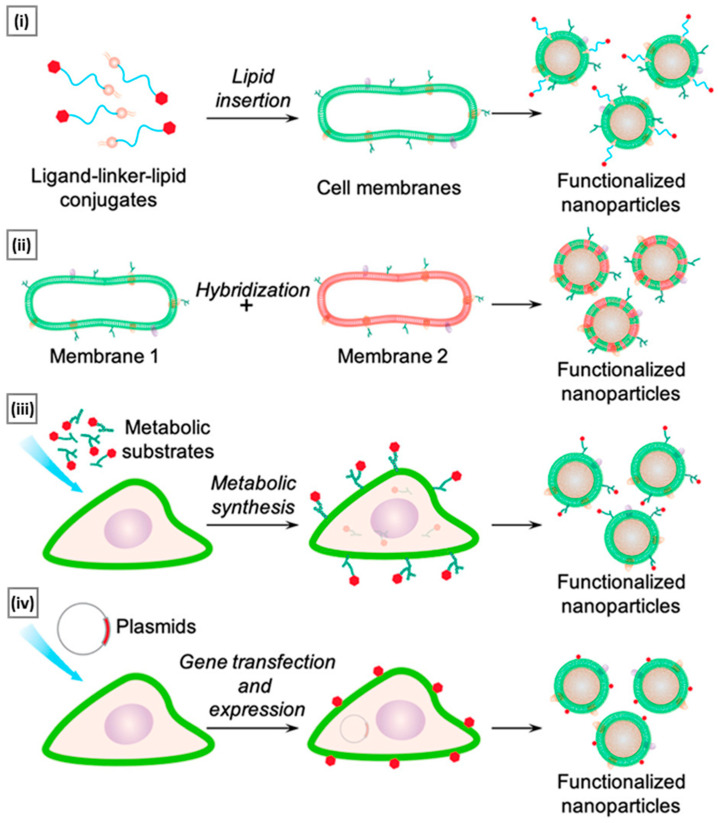
A schematic illustrating various approaches for functionalizing NPs coated with cell membranes: (**i**) lipid insertion, (**ii**) membrane hybridization, (**iii**) metabolic engineering, and (**iv**) genetic modification. Adapted with permission from [160], copyright American Chemical Society 2020.

**Figure 9 pharmaceutics-15-01677-f009:**
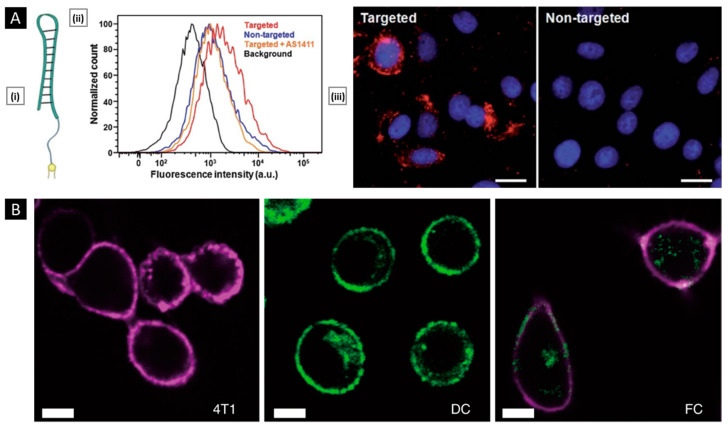
(**A**) (**i**) Schematic representation of AS1411-linker-lipid. (**ii**) Flow cytometry histograms of MCF-7 cells alone (black) and the cells incubated with AS1411-functionalized RBC-NPs (red), non-targeted RBC-NPs (blue), and AS1411-functionalized RBC-NPs together with the free AS1411 aptamer (orange). (**iii**) Fluorescence microscopy of MCF-7 cells incubated with AS1411-functionalized RBC-NPs and non-targeted RBC-NPs. DiD was loaded inside the RBC-NPs for visualization (red), and cellular nuclei were stained with DAPI (scale bars: 25 µm). Reproduced with permission from [165], copyright RSC Publishing 2013. (**B**) Fusion of DCs and 4T1 cells by CLSM observation over the red fluorescence of anti-CD44-APC antibody-marked 4T1, the green fluorescence of anti-MHC II-FITC antibody-labeled DCs, and the double-labeled FCs (scale bar: 10 μm). Reproduced with permission from [165], copyright Springer Nature 2019.

**Figure 15 pharmaceutics-15-01677-f015:**
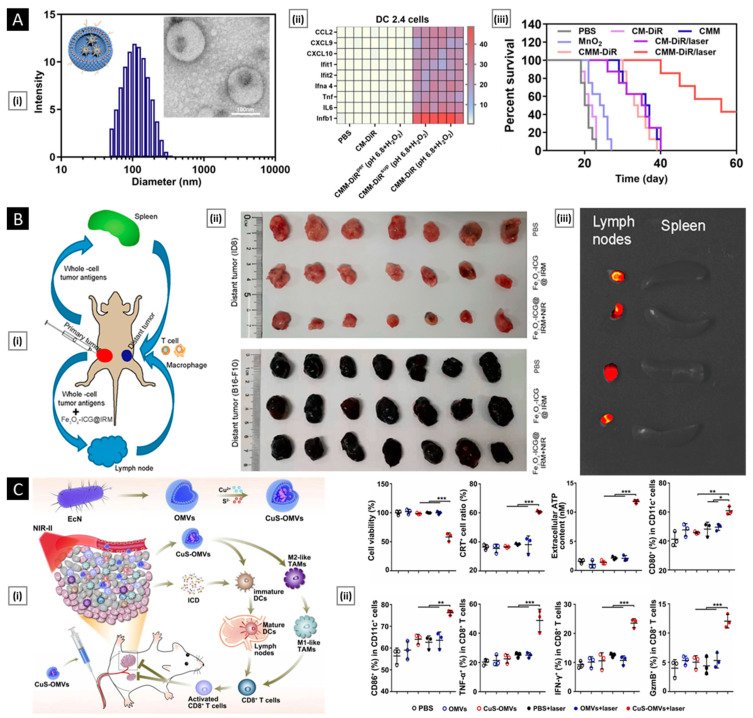
(**A**) (**i**) Size distribution histogram and TEM images of CMM-DiR. (**ii**) Relative gene expression of the cGAS-STING axis of DC 2.4 cells with different treatments for 24 h, detected by RT-qPCR analysis for evaluation of DCS maturation. (**iii**) The survival percentage of previously treated mice when re-challenged with tumor cells, showing a long-term immune memory effect. Reproduced with permission from [207], copyright Elsevier 2021. (**B**) (**i**) Schematic description of the mechanism of immunotherapy for metastatic tumors induced by Fe_3_O_4_-ICG@IRM. (**ii**) Distant tumor sizes of ID8 and B16-F10 in each group after treatment. (**iii**) Ex vivo fluorescence images of lymph nodes and spleens from C57BL/6 mice after intratumoral injection with Fe_3_O_4_-ICG@IRM (ICG = 8.5 μg) at 12 h. Reproduced with permission from [208], copyright American Chemical Society 2021. (**C**) (**i**) Schematic illustration of CuS-OMVs for synergistic PTT and immunotherapy in cancer treatment; (**ii**) Quantification of cellular parameters as indicators for CuS-OMVs-induced ICD, DC maturation, and T cell activation upon NIR-II light irradiation. (* *p* < 0.05, ** *p* < 0.01, *** *p* < 0.001). Reproduced with permission from [209], copyright Elsevier 2022.

**Table 2 pharmaceutics-15-01677-t002:** Examples of different membrane-coated nanoparticle cores.

Core	Delivery Cargo	Surface Modification	Application	Ref.
Blood Cells				
PEG nanogel	Doxorubicin (DOX)	-	Passive tumor targeting	[101]
Mesoporous silica	Graphene quantum dots; docetaxel	Cetuximab	PTT; light-triggered drug release	[102]
Manganese oxide	Chlorin e6; glucose oxidase	-	Tumor starvation; PDT	[103]
Nanocrystals	Docetaxel	cRGD peptide	Tumor targeted delivery	[104]
Silica	-	TNF-related apoptosis-inducing ligand	CTC-targeted delivery	[105]
Acetylated dextran	Bortezomib	Tissue plasminogen activator; alendronate	Bone-targeted delivery	[106]
Liposome	Emtansine	-	Metastatic tumor targeting	[31]
Iron oxide	-	TGF-β inhibitor (SB505124); anti-PD-1 antibody	Magnetically guided immune stimulation and ferroptosis	[107]
DSPE-PEG	IR-797	-	PPT; ICD-mediated immune stimulation	[108]
poly(lactic-co-glycolic acid) (PLGA)	Imiquimod	Anti-CD3 antibody	Local immune stimulation; T cell-directed tumor antigen presentation	[109]
PLGA	4,4′,4″,4‴-(Porphine-5,10,15,20- tetrayl) tetrakis(benzoic acid)	-	PDT; M1 macrophage polarization	[35]
Lipid NPs	Cyclic di-GMP	-	Immunogenic reprogramming of TME	[110]
MOF (PCN-224)	Glucose oxidase; catalase	-	Homotypic tumor-targeted PDT; starvation therapy	[111]
Iron oxide	-	Signal regulatory protein α	Magnetically guided immune stimulation; M1 macrophage polarization	[112]
PLGA	Imiquimod	Mannose	Antigen-presenting cell-targeted tumor antigen delivery	[113]
Gelatin nanogel	DOX	-	Tumor targeted delivery	[114]
Polydopamine NPs	SN-38	-	PTT; acidic TME-responsive drug delivery	[115]
MOF	DOX	-	Chemokine-mediated active targeting	[116]
Gold-iron oxide	Anti-miR-21	ICG	Homotypic tumor-targeted delivery, MR imaging	[117]
PLGA	DOX	Mesenchymal-epithelial transition factor binding peptide	Prolonged circulation lifetime; tumor-targeted delivery	[118]
Tyrosine-coupled dendrimers	let-7a mimics	-	Tumor targeted delivery	[119]
Gold-coated maghemite	-	-	MR imaging	[120]
Quantum dots	Anticancer drugs (DOX, cisplatin, and 5-fluorouracil) and/or siRNA cocktail and/or anticancer toxins	SP94 targeting peptide; histidine-rich fusogenic peptide (for endosomal escape)	Tumor targeted delivery, multimodal anticancer effect	[66]
β-cyclodextrin-modified GNPs and adamantane-modified Gold NPs	-	-	Inflammation-targeted PTT; ant-tumor immunotherapy; TME modulation	[121]
Titanium dioxide-coated magnesium Janus micromotor	-	-	Local tumor disruption; immune stimulation	[122]
Polyplex (PC7A)	CpG oligodeoxynucleotides (CpG 1826)	Maleimide	Tumor antigen capture and cross-presentation; immune stimulation	[84,85]

## Data Availability

Not applicable.

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
