# Peer review of "“Bioinspired” Membrane-Coated Nanosystems in Cancer Theranostics: A Comprehensive Review"

_pharmaceutics, 2023, doi:10.3390/pharmaceutics15061677_

Round 1
Reviewer 1 Report
In the manuscript entitled " "Bioinspired” Membrane-coated Nano-systems in Cancer Theranostics: A Comprehensive Review" the authors have described the potential of these nanosystems for precise image-guided cancer theranostics, covering various aspects such as sources of cell membrane, its isolation techniques, selection of nanoparticle cores, approaches of coating nanoparticle cores with cell membrane, and characterization methods. The authors have also highlighted the approaches for enhancing the multifunctionality of these nanosystems.
The article seems quite appropriate for the journal under consideration. The manuscript has been written and explained well and the text as well figure have been quite appropriately and well presented. The manuscript can well be accepted for publications in the journal. However, there are some concerns and suggestions which can be addressed before the final acceptance of the manuscript.
1. Introduction section appears quite short and brief and can be elaborated to further include other details and aspects.
2. The reviewer is unable to find any table in the manuscript. For better comprehension at least 2 tables should be incorporated in the manuscript, summarising the various cell types used as the cell membrane sources, and the other regarding the various nanomedicines and nanoparticles obtained from the cell sources along with the proper references.
3. A brief section summarising the various advantages and limitations of these cell types and cell membrane sources needs to be incorporated in the manuscript.
4. Conclusion and “future” perspective section seems overtly brief and simplistic and can further be expanded specifically to include more futuristic directions and aspects.
5. Various abbreviations have been used in the manuscript which should be expanded at the instance of their first use.
6. Some minor corrections and improvements in English language and grammatical errors need to be done in the manuscript.
e.g. line 68: centre etc.
Can be accepted after minor revision
Author Response
Reviewer 1
- Introduction section appears quite short and brief and can be elaborated to further include other details and aspects
Ans: As per the reviewer’s suggestion, introduction part has been elaborated and highlighted in yellow color.
- The reviewer is unable to find any table in the manuscript. For better comprehension at least 2 tables should be incorporated in the manuscript, summarizing the various cell types used as the cell membrane sources, and the other regarding the various nanomedicines and nanoparticles obtained from the cell sources along with the proper references.
Ans: As per the reviewer’s comment, one table summarizing the various cell membrane sources and their advantages as well as limitations have been included in the revised manuscript (Table 1). Various nanoformulations obtained from cell sources has also been tabulated in Table 2.
- A brief section summarizing the various advantages and limitations of these cell types and cell membrane sources needs to be incorporated into the manuscript.
Ans: As suggested by the reviewer, various advantages and limitations of several cell type and cell membrane sources has been included in the revised manuscript and highlighted in yellow color.
- Conclusion and “future” perspective section seems overtly brief and simplistic and can further be expanded specifically to include more futuristic directions and aspects.
Ans: As per the reviewer’s suggestion, conclusion and future perspective section has been expanded including more futuristic directions and aspects. Now we believe that it would fulfil the reviewer’s requirement.
- Various abbreviations have been used in the manuscript which should be expanded at the instance of their first use.
Ans: Required changes have been made in the revised manuscript.
- Some minor corrections and improvements in the English language and grammatical errors need to be done in the manuscript. [e.g., line 68: centre etc.]
Ans: Whole manuscript has been verified for minor corrections including grammatical errors.
Reviewer 2 Report
Title: “Bioinspired” Membrane-coated Nano-systems in Cancer Theranostics: A Comprehensive Review
Article Type: Review
Manuscript Number: pharmaceutics-2420655
Author: Desai N., Rana D., Pande S., et al.
In this comprehensive Review, AAs summarize and suggest the utility to use membrane-coated nano-systems as a promising approach for cancer theranostics.
In our opinion, the comments below must be addressed before considering the review acceptable for publication:
· 1. Multifunctionality to increase tumor active targeting is mandatory in our mind. AAs have to report some comparison data, in terms of tumor homing and anti-tumor effects in vivo, between “classical” targeted-nano-particles/liposomes and the membrane coated ones, in order to induce researchers/clinicians to choose the latter.
· 2. From a clinical point of view, how this novel platform can be managed? How this technology can be put into practice? Is the idea to withdraw blood cells, cancer cells, stem cells and so on from tumor patients, extract membranes, make membrane-coated nano-particles and re-infuse them into patients? In our opinion, it seems very hardworking, time consuming (especially for patients waiting for therapy) and very expensive, in other words, not easy to make. Please, add an exhaustive paragraph on this topic, highlighting related limitations of the proposed technology.
Author Response
Reviewer 2
- Multifunctionality to increase tumor active targeting is mandatory in our minds. AAs have to report some comparison data, in terms of tumor homing and anti-tumor effects in vivo, between “classical” targeted-nano-particles/liposomes and the membrane-coated ones, in order to induce researchers/clinicians to choose the latter.
Ans: We appreciate the reviewer’s keen observation. However, there is no study is available that quantitatively compares tumor targeting by membrane coating vs classic active targeting strategies. We have included relevant literature focusing on multifunctional advantages of membrane coated nanocarriers over classical nanoformulations. Further, comparisons with conventional systems have also been discussed in relevant sections of the revised manuscript.
- From a clinical point of view, how this novel platform can be managed? How this technology can be put into practice? Is the idea to withdraw blood cells, cancer cells, stem cells, and so on from tumor patients, extract membranes, make membrane-coated nano-particles, and re-infuse them into patients? In our opinion, it seems very hardworking, time-consuming (especially for patients waiting for therapy), and very expensive, in other words, not easy to make. Please, add an exhaustive paragraph on this topic, highlighting related limitations of the proposed technology.
Ans: We totally agree with the reviewer’s constructive comments on the clinical point of view. To bring this novel platform in actual clinical practice, it is utmost need to understand the challenges such as isolation techniques, translation of methodology and ethical approval. However, exhaustive clinical studies and technological advancement can make this novel platform to work in clinical settings.
Hence, the information highlighting the limitations of proposed technology has been included in the conclusion and perspective section of the revised manuscript.
Reviewer 3 Report
Topic of the manuscript is suitable for the pharmaceutics. Nevertheless, some point can be taken for its improvement.
1) Properties and usability of nanoparticles types should be introduced and compared in the table.
2) Discussed nanoparticles should be summarized in table/s.
3) Conclusion and perspective is to short and boring.
Author Response
Reviewer 3
- Properties and usability of nanoparticles types should be introduced and compared in the table.
Ans: As per the reviewer’s suggestion, properties and usability of nanoparticles types have been included in the tabular form in the revised manuscript. Additionally, core selection section has information regarding ideal core properties for coating.
- Discussed nanoparticles should be summarized in table/s.
Ans: Required tables have been included in the revised manuscript and highlighted in yellow color.
- Conclusion and perspective is to short and boring.
Ans: Conclusion and perspective part has been revised in more detailed in the revised manuscript and highlighted in yellow color. We believe that newly included information would fulfil the reviewer’s requirement.
Reviewer 4 Report
In this research, the authors reviewed "Bioinspired” Membrane-coated Nano-systems in Cancer Theranostics: A Comprehensive Review. Generally, it’s meaningful and interesting review. In my opinion, the current version of this manuscript fits the scope of Pharmaceutics and could be accepted after major revision.
My specific comments are in detail listed below:
1. In Figure 2, more types of delivery cargoes could be added including SDT drugs, radiotherapy drugs, or antibodies.
2. In Line 862-871, Line 872-890, and Line 1218-1260, how the immune system was affected by the "Bioinspired” membrane-coated nano-systems should be more clearly discussed. Some references should be added to this part including 10.1016/j.ijbiomac.2022.10.167.
3. Some minor mistakes exist in this review, such as CD8+ should be CD8+, Line 411 0.215±0.037 should be 0.215 ± 0.037. The authors should carefully check it.
4. In all the figures, the labels of all the small figures should be the same, such as A, B, C, D…
5. The advantage of using "Bioinspired” membrane-coated nano-systems should be more clearly pointed out when compared with the clinical usable liposome and albumin nanoparticles. Some references should be added to this part including 10.1016/j.jconrel.2022.11.004.
6. A more depth outlook or prospect that pointing out the clinical transformation possibility of the using the "Bioinspired” membrane-coated nano-systems should be added.
Author Response
Reviewer 4
- In Figure 2, more types of delivery cargoes could be added including SDT drugs, radiotherapy drugs, or antibodies.
Ans: Figure 2 has been modified with required changes as per the reviewer’s suggestion. Further, antibodies are the classic examples of biomolecules so they have been clubbed together in the figure.
- In Lines 862-871, Lines 872-890, and Lines 1218-1260, how the immune system was affected by the "Bioinspired” membrane-coated nano-systems should be more clearly discussed. Some references should be added to this part including 10.1016/j.ijbiomac.2022.10.167.
Ans: As suggested by the reviewer, required information has been discussed in the revised manuscript.
- Some minor mistakes exist in this review, such as CD8+ should be CD8+, Line 411 0.215±0.037 should be 0.215 ± 0.037. The authors should carefully check it.
Ans: Some minor mistakes have been corrected in the revised manuscript.
- In all the figures, the labels of all the small figures should be the same, such as A, B, C, D…
Ans: Required changes have been included in the revised manuscript.
- 5. The advantage of using "Bioinspired” membrane-coated nano-systems should be more clearly pointed out when compared with the clinically usable liposome and albumin nanoparticles. Some references should be added to this part including 10.1016/j.jconrel.2022.11.004.
Ans: As per the reviewer’s suggestion, advantageous of bioinspired membrane coated nanocarriers has been enlisted in tabular form in the revised manuscript.
- A more depth outlook or prospect that points out the clinical transformation possibility of using the "Bioinspired” membrane-coated nano-systems should be added.
Ans: Required information suggested by the reviewer has been included in the revised manuscript and highlighted in yellow color.
Round 2
Reviewer 2 Report
AAs clearly and exhaustively addressed the raised commments
Reviewer 3 Report
I have no objections
Reviewer 4 Report
The current version of this manuscript could be accepted.